# Exact inversion of partially coherent dynamical electron scattering for picometric structure retrieval

Benedikt Diederichs[1,2], Ziria Herdegen ●[1], Achim Strauch ●[3], Frank Filbir[2,4] & Knut Müller-Caspary ●[1,3] ✉

The greatly nonlinear diffraction of high-energy electron probes focused to subatomic diameters frustrates the direct inversion of ptychographic data sets to decipher the atomic structure. Several iterative algorithms have been proposed to yield atomically-resolved phase distributions within slices of a 3D specimen, corresponding to the scattering centers of the electron wave. By pixelwise phase retrieval, current approaches do not only involve orders of magnitude more free parameters than necessary, but also neglect essential details of scattering physics such as the atomistic nature of the specimen and thermal effects. Here, we introduce a parametrized, fully differentiable scheme employing neural network concepts which allows the inversion of ptychographic data by means of entirely physical quantities. Omnipresent thermal diffuse scattering in thick specimens is treated accurately using frozen phonons, and atom types, positions and partial coherence are accounted for in the inverse model as relativistic scattering theory demands. Our approach exploits 4D experimental data collected in an aberration-corrected momentum-resolved scanning transmission electron microscopy setup. Atom positions in a 20 nm thick $PbZr_{0.2}Ti_{0.8}O_3$ ferroelectric are measured with picometer precision, including the discrimination of different atom types and positions in mixed columns.

The spin-independent scattering of relativistic electrons by solids is governed by the Klein-Gordon equation. For a given stationary potential $V(\mathbf{r})$ of the target, and the incident electrons described by a pure state $\psi(\mathbf{r})$, the multislice approach[1] provides a coherent solution for calculating the scattered wave in the presence of dynamical scattering. It splits the specimen into slices and interweaves interaction with the projected potential of subsequent slices by Fresnel propagation. While simulating the scattering process for a known atomic potential landscape in forward direction is crucial, the true challenge lies in solving the problem of phaseless inverse multiple scattering, i.e.,

the retrieval of physical parameters characterizing the specimen and the illumination from recorded intensities. Considering contemporary nanostructures in the field of information or energy technology, their functionality is determined by the structure-property relationship at the atomic level. Prominent specimen parameters to be measured in preferably direct manner are, therefore, atomic positions, species, and thermal vibrations.

The rise of 4D scanning transmission electron microscopy (STEM), in which real and momentum space information are simultaneously available by recording a full diffraction pattern for each

[1]Department of Chemistry and Centre for NanoScience, Ludwig-Maximilians-Universität München, Munich, Germany. [2]Institute of Biological and Medical Imaging, Helmholtz Zentrum München, Neuherberg, Germany. [3]Ernst Ruska-Centre for Microscopy and Spectroscopy with Electrons, Forschungszentrum Jülich, Jülich, Germany. [4]Department of Mathematics, TUM School of Computation, Information and Technology, Technische Universität München, Garching, Germany. ✉e-mail: k.mueller-caspary@cup.lmu.de

position of the electron probe, is presently accompanied with the development of advanced reconstruction methods, most notably gradient-based schemes[2–5]. They enabled the retrieval of individual slice transmission functions at spatial resolutions in the range of thermal disorder, and the recovery of the incident probe[6]. Importantly, contemporary inversion schemes reconstruct both multimodal illumination[6–9] and unimodal specimen. Both the potential of the specimen as well as the states of the illumination, are discretized on a fine grid as pixelated images. These complex-valued images are then optimized to achieve numerical consistency with the experimental observations. While multimodal specimens have been proposed[7,10] and applied to optical experiments neglecting multiple scattering[11], working out the conceptual framework for partially coherent inverse multiple electron scattering and the application to experimental data remains as a central challenge. Additionally, the pixelwise inversion strategy is not confined to yield physically plausible Coulomb potentials $V(\mathbf{r})$ and wave functions $\psi(\mathbf{r})$. Furthermore, any atomistic interpretation has to be inferred *a posteriori*, by means of image processing being independent of the experimental data.

This study first works out an inversion scheme whose solutions for the potentials and wave functions inherently satisfy physics laws. In particular, it resolves contemporary model violations that occur because both the specimen and the illumination are in mixed states due to, inter alia, thermal disorder and partial temporal and spatial coherence of the incoming electron beam. Each state satisfies the Klein-Gordon equation individually, and all states are in incoherent superposition. This makes their recovery from phaseless inverse scattering data significantly harder, since a multitude of multislice cycles with a variety of thermal ensembles of the potentials and different incoming wave functions needs to be inverted.

To this end, a method is proposed in which the entire mixed-state scattering problem is comprehensively parametrized by physical quantities in both forward and inverse direction. It uses a few hundred free parameters, among them the positions, types and thermal displacements of atoms as well as probe aberrations. This reduces the number of unknowns by at least four orders of magnitude as compared to established methods while yielding wave functions and potentials fully consistent with quantum mechanics and electrostatics, respectively. Via efficient gradient calculation employing an exact neural network representation of dynamical scattering[12], it is demonstrated that even the complexity of fully incoherent thermal diffuse scattering (TDS) can be disentangled in inverse direction. This provides drastically enhanced chemical sensitivity due to atomic number contrast dominating TDS at high scattering angles, and also introduces the temperature of the specimen as a differentiable parameter that becomes a direct observable by inverting the four-dimensional data. Besides meeting physics constraints exactly, our fully atomistic parametrization of the partially coherent inverse problem requires no handcrafted regularization strategies that are commonly needed to stabilize existing algorithms. Furthermore, parameters of interest are directly available, making additional processing steps redundant.

Our method is developed using both experimental and simulated data. It is demonstrated that symmetry breaking in the atomic structure of ferroelectrics is accurately retrieved from a coarse, fully symmetric initial guess, with a precision of 3 pm for the atomic positions. Determining the local chemical composition by analysing the gradient of occupancies for mixed atomic columns is outlined, and found accurate only for solving the inverse problem with TDS. Most importantly, ferroelectric displacements in $PbZr_{0.2}Ti_{0.8}O_3$ are determined in simulations and 4D-STEM experiments, enabling the separation of atomic sites below the Rayleigh limit applied to the projected potential.

## Results

### Exact inversion of partially coherent phaseless scattering data

The frozen phonon (FP) approach translates time-dependent potentials $V(\mathbf{r}, t)$ arising from the thermal motion of atoms, to an incoherent superposition of different stationary states $V_i(\mathbf{r})$. Figure 1a exemplifies this concept via the projected potentials of two slices $V_{FP}^{(1,2)}(\mathbf{r})$ in a ferroelectric $PbZr_{0.2}Ti_{0.8}O_3$ (PZT) crystal. The potentials contain thermal disorder via statistically perturbing the ideal atomic positions of the crystal by thermal displacements $\mathbf{u}$, and thereby breaking the symmetry, which leads to TDS in diffraction space. Experimentally observable signals are then obtained by averaging over many of these displacement configurations[13,14]. Consequently, there exists no single potential landscape that is responsible for the measured intensities. The significance of TDS is demonstrated in Fig. 1b-e by means of an experimental momentum-resolved 4D STEM data set recorded at in-focus conditions using an aberration-corrected microscope. The

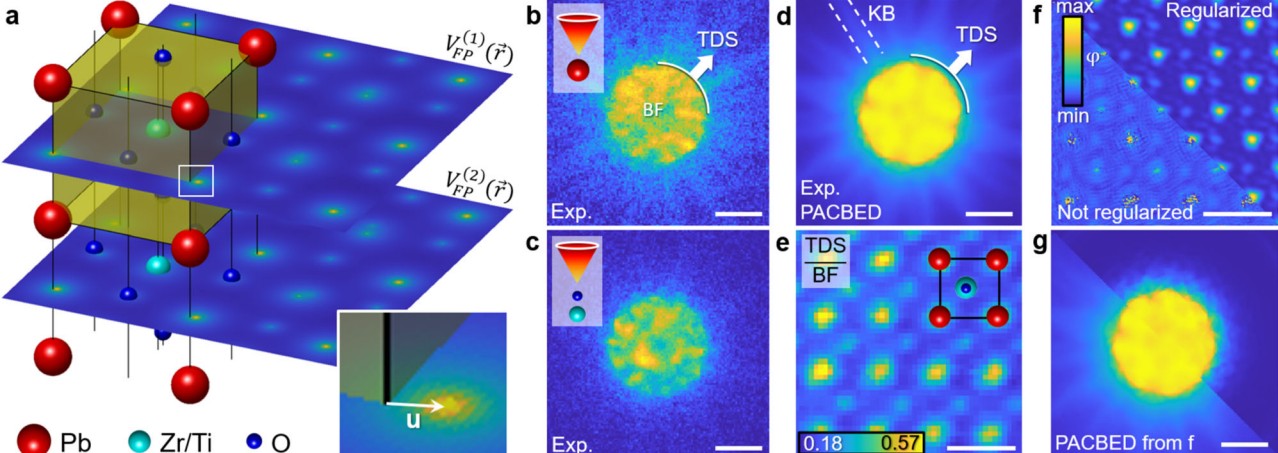

**Fig. 1 | Phaseless inverse problem with thermal disorder in $PbZr_{0.2}Ti_{0.8}O_3$.**
**a** Schematic multislice scheme with 2 slices $V_{FP}^{(1,2)}(\mathbf{r})$ of a single configuration $\tau$ and thermal displacements. **b, c** Exp. diffraction patterns with the aberration-corrected scanning transmission electron microscopy (STEM) probe on **b**: Pb, **c**: ZrTiO column showing different amount of thermal diffuse scattering (TDS). **d** Position-averaged convergent-beam electron diffraction (PACBED) pattern for an exp. 4D STEM scan across $5 \times 5$ unit cells in PbZrTiO₃, with Kikuchi bands (KB) formed by TDS. **e** Ratio of TDS-dominated dark field intensity to the bright field (BF) intensity, reaching 60% at Pb. **f** Pixelwise reconstruction of the slice transmission function (phase). Interstitial regions and atom shapes contain artefacts arising from disorder not accounted for in the reconstruction model. **g** Simulated PACBED using the phase grating from **f**. The pixelwise reconstruction created diffuse intensity as an artefact to numerically match the experiment from **d**. Scale bars **b–d**, **g**, 25 mrad. Scale bars **e**, **f**, 5 Å.

bright field (BF) region is a complex interference pattern formed by the primary beam and Bragg reflections. In thicker specimens the dark field is dominated by TDS, which itself exhibits a fine structure in the form of Kikuchi Bands (KB), see Fig. 1d. The ratio of TDS and BF intensity in Fig. 1e constitutes that TDS scales up to approximately 60% at Pb columns. It is therefore essential that TDS be included when retrieving details of the scatterer by comparing diffraction patterns that include parts of the dark field.

Established inversion concepts determine the phase $\varphi(\mathbf{r})$ of slice transmission functions $A(\mathbf{r}) \cdot \exp[i\varphi(\mathbf{r})]$ as pixelated images, treating the values of neighboring pixels as independent. The result of this approach is depicted by the pixelwise inverse multislice result in Fig. 1f, where a number of 21 slices has been used with a Fresnel distance of 7.9 Å, corresponding to two unit cells in electron beam direction. While partial coherence of the illumination was included, a unimodal specimen was reconstructed without (bottom left) and with regularization to enforce $\varphi(\mathbf{r})$ to be continuous (top right). The atomic structure is clearly visible, including distinguishable heavy (Pb), medium (ZrTiO) and light (O) columns. However, the phase distribution contains a variety of artefacts both at the atomic sites and in the interstitial region. Their origin becomes apparent in Fig. 1g, where the reconstructed transmission functions from f were used to calculate the reconstruction-based 4D STEM data. In Fig. 1g, this data has been compacted to the scan position-averaged convergent-beam electron diffraction pattern (PACBED). Although it is physically impossible to produce TDS without any ensemble averaging, the non-regularized result in Fig. 1g exhibits a nearly 1:1 agreement with the experiment in Fig. 1d. This provides rather numerical consistency with the measurement than a physical representation of the scatterer, since it is neither required that $\varphi$ resembles a physical potential $V$ nor can it include thermal disorder due to the single-state treatment of the specimen. A detailed simulation study is shown in Supplementary Fig. 1. On the other hand, regularization strategies can assure smooth and continuous phases top right in Fig. 1f, but are incapable of producing the details of thermal diffuse scattering in diffraction space as seen from the corresponding PACBED top right in Fig. 1g. Though the quality of pixelwise reconstructed phases depends on acquisition settings and regularization details, explicitly accounting for phonons in the solution to the inverse problem is crucial to exploit the entire information in 4D-STEM experiments to include especially high-angle scattering.

We resolve this model violation by requiring $\varphi = \sigma \cdot V_{\mathrm{FP}}(\mathbf{r})$ with the interaction constant $\sigma$ and a Coulomb potential $V_{\mathrm{FP}}(\mathbf{r})$ produced by a number of atoms. Let $v_Z$ be the projected Coulomb potential of an atom with atomic number $Z$, which is obtained from Hartree-Fock calculations for isolated atoms via contemporary parametrizations[15]. Assuming that the $j$th slice contains $N$ atoms with atomic numbers $Z_n$, equilibrium positions $\mathbf{r}_n \in \mathbb{R}^2$ and weights $w_n$, $n = 1, ..., N$, the total potential of the slice for a single thermal configuration $\tau$ is given by

$$V_{\mathrm{FP}}^{(j)}(\mathbf{r}, \tau) = \sum_{n=1}^{N} w_n v_{Z_n}\left(\mathbf{r} - \mathbf{r}_n - \sqrt{\langle u_n^2 \rangle} \cdot \mathbf{g}_{n,\tau}\right), \qquad (1)$$

where $\langle u_n^2 \rangle$ is the mean squared thermal displacement. The atomic potential is thus explicitly shifted according to the frozen phonon model to introduce thermal disorder. To this end, $N$ independent and identically distributed two dimensional normal Gaussian variables $\mathbf{g}_{n,\tau}$ are drawn and $V_{\mathrm{FP}}^{(j)}$ is recalculated every time the slice is used. Optimizing mean and standard deviation of a probability distribution in this way is a standard approach when optimizing variational autoencoders[16]. Anisotropic thermal vibrations can be incorporated similarly. In known systems, we let $w_n = 1$. However, these weights can be optimized to detect an incorrect atomic number or a missing atom. Since it can be computationally demanding to explicitly account for TDS during the inversion, temperature effects can also be taken into account via the Debye-Waller damping scheme. In that case, $\mathbf{g}_{n,\tau} = 0$ in

Eq. (1), whereas potentials $v_Z$ are adjusted for the Debye-Waller factor and absorptive potential[17]. The latter treats TDS as an absorption and allows for a reconstruction that is mostly based on the bright field region.

We then describe the interaction with the specimen for one frozen phonon state $\tau$ by the multislice operator $\hat{M}_{\{S\}}(\tau)$, which stands for a series of slices having potentials according to Eq. (1), and the convolution with the Fresnel propagator in between. The set $\{S\}$ includes all parameters that describe the specimen. Accordingly, we succinctly express the illumination at scan point $\mathbf{s}$ by $(\alpha_k \phi_{\{P\}}(\mathbf{r} - \mathbf{s} - \boldsymbol{\eta}_k, \delta_k))_k$. The set $\{P\}$ includes all parameters of probe formation such as the size of the probe-forming aperture and aberration coefficients, $\boldsymbol{\eta}_k$ reflects a disturbance of the actual scan position due to partial spatial coherence. $\delta_k$ accounts for partial temporal coherence in terms of focus fluctuations, and $\alpha_k$ measures the probability that a wave with $(\boldsymbol{\eta}_k, \delta_k)$ is present in the illumination. The diffracted intensity at scan coordinate $\mathbf{s}$ parametrized according to contemporary scattering theory in the presence of TDS and finite coherence then reads

$$
\begin{aligned}
I(\mathbf{s}) &= \mathbb{E}_{\tau, \delta, \eta}\left[\left|\hat{M}_{\{S\}}(\tau)\phi_{\{P\}}(\mathbf{r} - \mathbf{s} - \boldsymbol{\eta}, \delta)\right|^2\right] \\
&\approx \frac{1}{N_\tau} \sum_{k,\tau} \alpha_k \left|\hat{M}_{\{S\}}(\tau) \cdot \phi_{\{P\}}(\mathbf{r} - \mathbf{s} - \boldsymbol{\eta}_k, \delta_k)\right|^2 \qquad (2) \\
&\approx \sum_k \alpha_k \left|\hat{M}_{\{S\}}(\tau_k) \cdot \phi_{\{P\}}(\mathbf{r} - \mathbf{s} - \boldsymbol{\eta}_k, \delta_k)\right|^2 .
\end{aligned}
$$

The approximation ties the summation over thermal ensembles into the incoherent superposition of different probe wave functions, each of which interacting with a new FP configuration $\tau$. Note that while the coherence parameters are optimized and assumed to be the same for each scan point, $\tau$ is re-rolled in each step of the optimization.

One can now define a loss function between experimental diffraction patterns as in Fig. 1b, c and $I(\mathbf{s})$ as a whole, i.e., including TDS. This poses the challenge to determine unknown parameters in $\{S\}$ and $\{P\}$ that minimize the loss, typically considering $10^4 - 10^5$ diffraction patterns recorded in a 4D STEM experiment. Most importantly, an efficient concept is needed to compute the gradients with respect to all physical parameters of interest. Explicitly determining the loss and evaluating the required gradients can then be performed in a single step, such that updates for $\{S\}$ and $\{P\}$ are immediately obtained. In the present work, this is achieved by recasting the multislice function as an artificial neural network, following an idea put forward in Refs. 2,3, implemented in Pytorch[18]. That allows utilizing automatic differentiation (AD) schemes, capable to seamlessly calculate the gradients of all relevant parameters, as has been utilized for multimodal pixelwise X-ray ptychography[19]. For details we refer to the Methods section and Supplementary Notes 1–4.

In addition to its consistently physical formulation, our approach reduces the complexity of the reconstruction problem. The number of unknown parameters drops from determining typically $10^6$ complex pixel values by four orders of magnitude to usually a few hundred real parameters in $\{S\}$ and $\{P\}$. Our approach operates without any regularization, eliminating empirical and thus case-dependent and partly ambiguous regularization concepts[5,6], which are always necessary in a highly over-parameterized regime and commonly used.

## Atomic structure retrieval in theory

As to the application, a long-standing challenge in nanoscience consists in quantifying the local polarization in ferroelectrics. For example, a permanent dipole moment is produced in tetragonal $PbZr_{0.2}Ti_{0.8}O_3$ by ionic displacements of the pure oxygen columns, and by the atoms in the (ZrTi)O column as already illustrated in the inset in Fig. 1f. Whereas the interpretability of phase-contrast imaging[20,21] concerning polarization-induced electric fields in materials with broken Friedel symmetry can suffer significantly from dynamical diffraction,

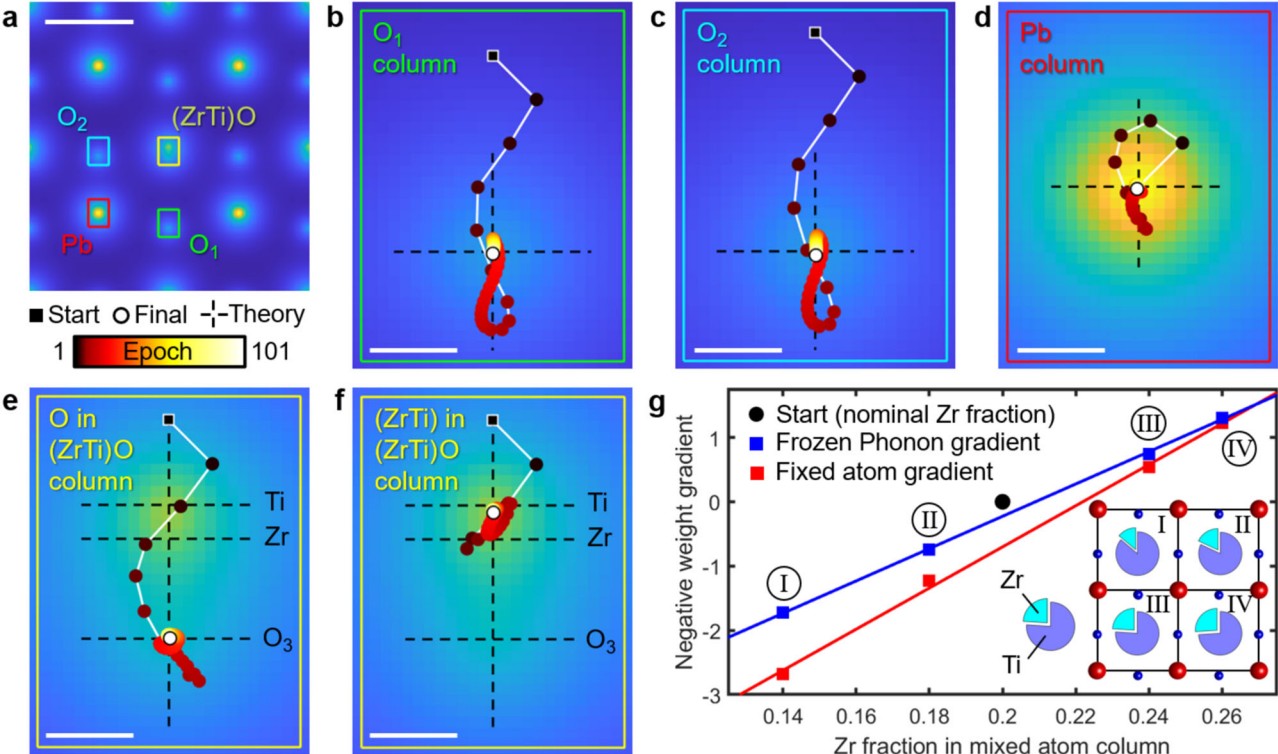

**Fig. 2 | Simulation study of recovering atomic positions in ferroelectric PbZr$_{0.2}$Ti$_{0.8}$O$_3$. a** Reconstructed potential distribution, ferroelectric displacements are identified by eye. For the four atomic columns (colored rectangles), **b–f** show magnified views of the reconstruction pathways vs. epoch. Scale bar, 2.5 Å. **b–f** Reconstruction pathways based on atom position gradients for all sites with start (black square, symmetry position), final (white circle) and theoretical (dashed) position. The rectangular patches **b–f** represent magnified regions in **a** with the respective color. Parts **e**, **f** show that even though O and (ZrTi) started exactly at the same position, the algorithm is able to pick them apart. Scale bars, 20 pm. **g** Atomic weight gradients to identify chemical composition in (ZrTi) with different Zr fractions using frozen phonon (FP) and Debye-Waller based inverse models. The gradient for the initial models with 20% Zr is linear and adopts nearly zero at 20% only for the FP case.

resolving the ferroelectric displacements of atoms directly[22–24] appears more suitable. However, the slightest crystallographic mistilts and aberrations hamper quantitative picometer-scale structure retrieval by conventional methods drastically[25], especially since adjacent domains are often tilted with respect to each other[21]. Consequently, quantifying the ferroelectric displacements in tetragonal PbZr$_{0.2}$Ti$_{0.8}$O$_3$ is considered an ideal application in the following to disentangle illumination, scattering dynamics, and atomic structure via physically parametrized inversion of 4D STEM data including TDS and partial coherence.

The methodological concept to retrieve structure and the chemical composition of the ZrTi site is elaborated via simulated data in Fig. 2. 4D STEM diffraction patterns have been simulated with TDS in FP mode, and partial spatial and temporal coherence for the illumination was included. Ferroelectric displacements of PbZr$_{0.2}$Ti$_{0.8}$O$_3$ have been taken from literature[26], whereas the Zr and Ti atoms were distributed randomly according to a Zr fraction of 0.2. Further simulation details are given in the Methods section. In practice, the polarization is unknown, thus specimen parameters {S} for the initial reconstruction were set up without ferroelectric displacements. For the ZrTi site, the virtual atom approximation was employed for the reconstruction where the potential is formed by a linear combination of 0.8 parts Ti and 0.2 parts Zr.

Figure 2 a depicts the Coulomb potential distribution corresponding to the atomic structure in the reconstructed slice. With Pb marking the corners of the tetragonal unit cell, the rather significant ferroelectric shifts of the pure oxygen columns are clearly seen by the eye. Notably, the mixed (ZrTi)O column is reconstructed asymmetrically, indicating that the reconstruction is able to distinguish the slightly different oxygen position with respect to the (ZrTi) site. The potential distribution of the initial model is shown in Supplementary Fig. 2.

To shed light on the reconstruction progress, magnified pathways of each atom column are mapped in Fig. 2b–f in dependence of the iteration epoch. The black square marks the start coordinate (symmetry position) and the white circle the final atom position, with the course in between color-coded by epoch. In all cases, the ground truth as labeled by the dashed black cross, is retrieved back perfectly. This validates the treatment of the (Zr$_{0.2}$Ti$_{0.8}$) site by means of a virtual atom during the inversion process, which is further supported by Fig. 2f, where the final position is given by 0.2 and 0.8 times the ferroelectric displacement of Zr and Ti, respectively. Even the oxygen atom O$_3$ in Fig. 2e, located in the same atomic column, adopts its correct position. Note the fundamental difference to non-parametrized inversion schemes, where the reconstruction would, in the ideal case, yield an asymmetric phase distribution around the (ZrTi)O column, deduced from pixelwise calculated gradients without direct physical interpretability. Here, Fig. 2b–g reflects gradients and final values of ten explicit physical parameters, that is, $x$ and $y$ coordinates of the shown atom columns. Further reconstruction details are given in Methods section.

The greatly increased sensitivity of our inversion method to the chemical composition via Z-contrast, arising from the exact inclusion of high-angle TDS, is shown in Fig. 2g. 4D-STEM data have been simulated for a random alloy (Zr$_x$Ti$_{1-x}$) as before, except that the composition $x$ fluctuated statistically, as depicted for four selected columns with Zr fractions $x$ between 14 and 26%. In practice, only the average Zr fraction of 20% may be known which we used to initialize

the parametric model {S} with a virtual $(Zr_{0.2}Ti_{0.8})$ atom. Its potential is approximated by the linear combination of 20% the potential of Zr and 80% Ti, corresponding to an effective atomic number of $Z_{eff} = 25.6$. None of the mixed columns I-IV used to simulate the diffraction patterns adopts this composition exactly. Consequently, the inversion procedure captures this discrepancy by detecting too strong scattering potentials, or equivalently, too high weights $w_n$ in Eq. (1) in the model at (ZrTi) sites I and II, and vice versa at sites III and IV. For sites I-IV, the negative gradient of the loss with respect to the weights was

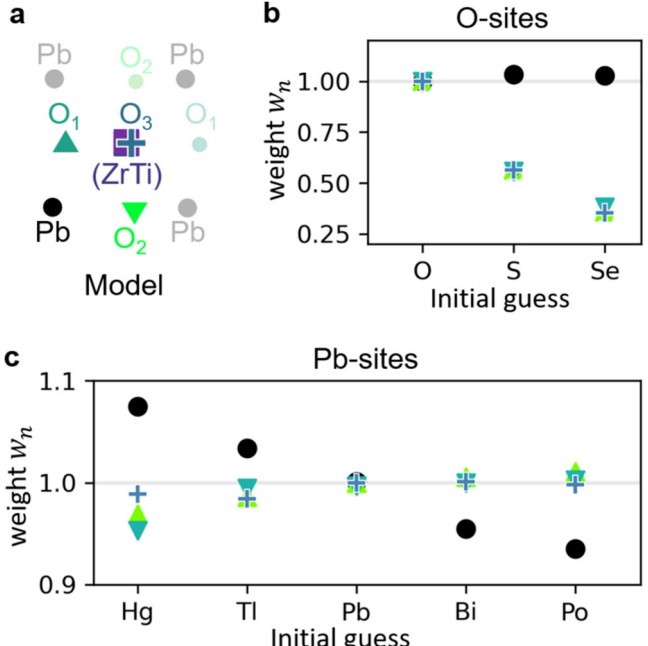

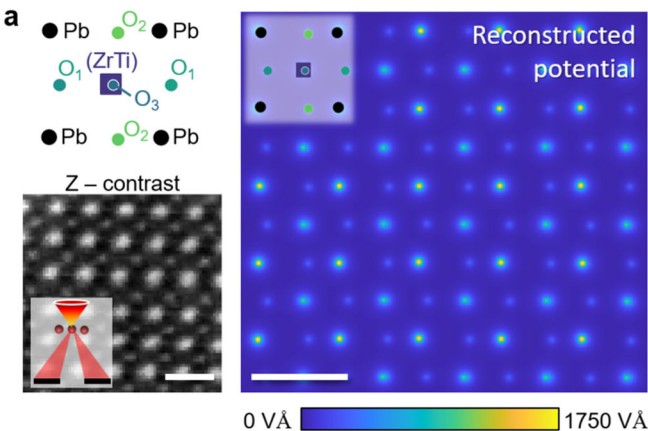

**Fig. 3 | Detection of correct atom types.** Simulation of 4D scanning transmission electron microscopy (STEM) data for the model in **a** has been performed using frozen phonons. The initial model has been set up by placing **b** oxygen and **c** lead sites with different atom types. Shown are the optimized weights after 20 epochs employing an inverse model with 10 frozen phonon configurations. The weights clearly indicate O and Pb as the correct solution, and suggest lowering or increasing the atomic number for too heavy or light atoms, respectively.

determined using an inverse model that includes TDS via frozen phonons (blue), or neglects thermal disorder within the Debye-Waller approach (red), respectively. In both cases, a clear linear relation can be seen, whereas only the FP gradient nearly vanishes in case the model adopts already the true composition (black dot in Fig. 2g). Neglecting TDS (red) in the inverse multislice scheme would thus lead to the wrong chemical composition of the $(Zr_{0.2}Ti_{0.8})$ column. This study not only demonstrates that a misestimated chemistry in an initial parametrization of {S} in mixed columns can be corrected, it also confirms that treating (ZrTi) as a virtual atom is justified here which will be utilized in the experimental study below.

We close the theoretical study by investigating the capabilities of the parametric approach to detect the correct atomic types, since they might not be known exactly in practice. To this end, we optimized the weights $w_n$ in Eq. (1) for initial models in which we substituted either lead or oxygen by the atomic species shown in Fig. 3b, c. Indeed, the graphs show that $w_n = 1$ is obtained for the correct occupation with O and Pb according to the $PbZr_{0.2}Ti_{0.8}O_3$ ground truth. Importantly, erroneously starting with a PbS or PbSe host crystal as in Fig. 3b suggests to weigh the S and Se potential by approximately 50 and 35%, respectively. Note that also the Pb weights have been optimized and remain close to one. Similarly, a fictitious substitution of Pb by Hg, Tl, Bi or Po shown in Fig. 3c yields an almost linearly adapted increase or decrease of the scattering potential for too light or too heavy atoms, respectively. While substitution at the Pb site changes also the oxygen weights slightly by less than 5%, a confusion with S or Se can be ruled out according to Fig. 3b.

## Inversion of scattering from a ferroelectric

The applicability of our approach is ultimately demonstrated by the inversion of experimental momentum-resolved STEM data of 22 nm thick ferroelectric $PbZr_{0.2}Ti_{0.8}O_3$ in Fig. 4. The inversion strategy is based on the simulation study in conjunction with Fig. 2. Importantly, the Z-contrast image of the reconstructed region in Fig. 4a declares accurate structure retrieval by established techniques futile, given the drastic distortions, and the complete absence of oxygen. Instead, not only the gradients of the atomic parameters, but also those of the raster positions, aberrations and partial coherence have been minimized within the 52-slice inversion scheme, encapsulated in a complex inverse model involving TDS via thirteen frozen phonon configurations. Atom types and their reconstructed equilibrium positions are

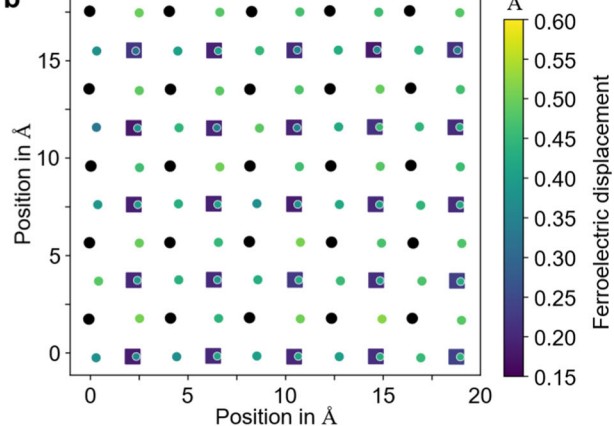

**Fig. 4 | Experimental polarization measurement in ferroelectric $PbZr_{0.2}Ti_{0.8}O_3$ by parametric diffraction inversion. a** Scanning transmission electron microscopy (STEM) Z-contrast image generated from experimental 4D STEM data showing only the heavy atomic columns, Pb and (ZrTi)O₃, and the strongly distorted scan grid due to drift and irregular scan raster. Ferroelectric displacements are seen directly in the potential map (color-coded), which was generated from the

atomic structure reconstructed taking TDS and partial coherence into account. Scale bar, 5 Å. **b** Map of reconstructed atomic column positions with different sites distinguished by symbol. Pb atoms (black) were used as a reference frame to calculate the ferroelectric displacements from the symmetry position (color-coded). The map shows homogeneous polarization with consistently determined atom positions, including the separation of (ZrTi) and O in the mixed column.

subject to the potential distribution in Fig. 4a. The rightward ferro-electric polarization is seen by eye. Note that all atom coordinates in the plot have been reconstructed individually without any prior polarization. In that respect, the excellent structural fidelity of the reconstruction is achieved by the parametrization paradigm, whose self-calibrating nature arises from the required compatibility of the slice transmission functions in the inverted scattering model with physical Coulomb potentials. In other words, constraining the slice potential to a linear combination of known atomic ones locks the relative positions of the scan, and by that virtue, the neighboring atomic positions. Here, the reconstructed lattice parameters of 0.3932 and 0.4121 nm (polar direction) and their ratio of 1.05 agree excellently with earlier reports[26]. A direct comparison of experimental raw data and corresponding diffraction patterns from the reconstruction is presented in Supplementary Fig. 3.

A detailed quantitative structural view on the measured atomic-scale PZT ferroelectricity is presented in Fig. 4b. With each four Pb atoms (black) defining a local reference cell, the internal coordinates of all sites have been determined and their deviation from the non-ferroelectric symmetry positions mapped color-coded, symbols labeling atom types. The homogeneities of the ferroelectric displace-ments of the individual sites (ZrTi) and $O_{1,2,3}$ as expressed by the respective nearly monochrome symbols throughout the field of view, is striking. In particular, the asymmetric unit (ZrTi)-$O_3$ appears con-sistent across all unit cells, verifying the capability to decipher sites of different atoms in a single column experimentally. This is further underpinned by the statistical evaluation presented in Table 1, which contains the average displacement over the 25 unit cells covered completely by the scan. Overall, a very good agreement with literature[26] is found. Only $O_3$ deviates by 9 pm, a reasonable result as this oxygen shares the atomic column with ZrTi.

### Table 1 | Measured ferroelectric displacements in PbZr$_{0.2}$Ti$_{0.8}$O$_3$

| Site | c direction | | a direction | |
|---|---|---|---|---|
| | Exp. [pm] | Lit. [pm] | Exp. [pm] | Lit. [pm] |
| Ti/Zr | 20 (2) | 17/26 | 0 (2) | 4˙0 |
| $O_1$ | 42 (4) | 43 | 2 (3) | |
| $O_2$ | 48 (2) | 48 | 5 (3) | |
| $O_3$ | 39 (4) | 48 | 1 (2) | |

Coordinates of atoms from the parametric inversion of 4D STEM data in Fig. 4b were statistically evaluated across 25 unit cells in the reconstructed region. The standard deviation is given in brackets, literature references have been taken from Ref. 26.

Finally, Fig. 5a depicts one of the reconstructed quantum mechanical wave functions for the illuminating probe (modulus), with its Fourier transform as inset (phase) as determined from the aberra-tion coefficients. It adopts the physically realistic shape for slightly astigmatic in-focus conditions. The initial (black) and reconstructed (red) partial spatial coherence effects are expressed in Fig. 5b where the weights $\alpha_k$ are shown relative to $\|\boldsymbol{\eta}_k\|$. Starting from a Gaussian model, the reconstructed weight distribution is more heavily tailed, which agrees perfectly with dedicated studies[27] proposing a bivariate Cauchy distribution added as a fit (red). Details are given in Supple-mentary Note 5.

## Discussion

The proposed approach to the inverse phaseless dynamical scattering problem changes the reconstruction paradigm to allow solely physical solutions for the Coulomb potential and quantum mechanical wave functions. Simultaneously, incoherent thermal diffuse scattering so far inaccessible by scattering matrix[28] or inverse multislice methods is incorporated exactly, the number of unknowns is reduced by four orders of magnitude, the inversion procedure is self-calibrating in real space and renders regularization obsolete. However, while the pro-posed direct parametrization of coherence effects can be always used, the optimization of atomic parameters requires a reasonable initial model. That can be constructed from a pixelated phase grating. Our findings on picometer-level ferroelectric structure retrieval in PbZr$_{0.2}$Ti$_{0.8}$O$_3$ as to unit cell geometry and chemical composition open promising perspectives to decipher structure-property relationships determining the physics of nanostructures. The availability of the gradients for all relevant parameters especially opens another inter-esting future research direction to optimize experimental design. By simulation, one could identify those experimental conditions which are most sensitive to, e.g., substituted atoms (Fig. 2g), vacancies or any other parameter of interest. Ultimately, including chemical bonding either by modified atomic scattering amplitudes[29,30] in the parametric model, or by augmenting the model with a pixelwise reconstructed charge density difference map renders direct access to valence charge redistributions feasible, being central for catalytic and transport properties.

## Methods
### Details of inversion scheme
Our implementation relies on Pytorch and its automatic differentiation techniques. Derivatives of complex-valued functions are thus calcu-lated using Wirtinger calculus. For each type of interaction and pro-pagation a layer was implemented. These layers can be stacked

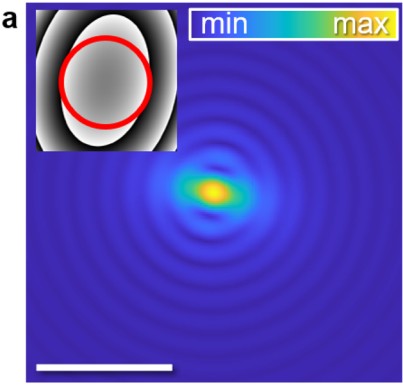
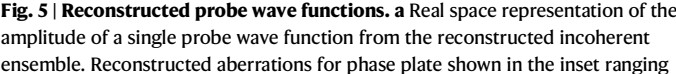
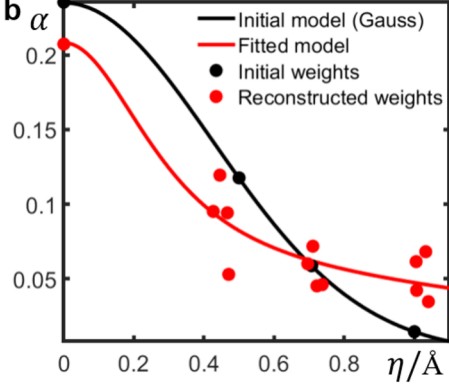

**Fig. 5 | Reconstructed probe wave functions. a** Real space representation of the amplitude of a single probe wave function from the reconstructed incoherent ensemble. Reconstructed aberrations for phase plate shown in the inset ranging from $-\pi$ (black) to $\pi$ (white): $A_1 = 13.58$ Å (204.12˚), $A_2 = 9.51$ Å (128.41˚), $B_2 = 8.67$ Å (126.7˚). Scale bar, 3 Å. **b** Initial (black) and reconstructed (red) weights $\alpha_k$ of incoherently superimposed probes versus source coordinate $\eta_k$.

together arbitrarily, including repetitions. That allows for models that are periodic in beam direction. Any such model is then represented by a neural network and can be "trained", i.e., fitted to experimental data, by one of the optimizers provided by Pytorch. We mostly rely on Adam or a standard stochastic gradient descent. It is stressed that we use a holistic approach, where all parameters are optimized simultaneously. As the computational graph, used in the backpropagation algorithm for calculating the gradients, has to be build anyways, adding some more nodes for scan positions and other parameters adds only little complexity. We use different learning rate schedulers to activate the adjustments of scan positions, aberration parameters and coherence parameters after a few epochs.

The frozen phonon layer is an implementation of Eq. (1). For reasons of efficiency, $v_z$ is precalculated using a parametrized potential[15]. To allow for subpixel shifts with a precalculated potential, the addition is carried out in the frequency domain. Equation (1) is for clarity written for isotropic thermal vibrations. For each atomic site, these are characterized by the mean squared thermal displacements $\langle u_n^2 \rangle$ along one dimension, relating to the isotropic Debye parameter via $B_n = 8\pi^2 \langle u_n^2 \rangle$ and to the Debye-Waller factor as $\exp(-\frac{1}{4}Bq^2)$ with spatial frequency $q$. For anisotropic cases, $\sqrt{\langle u_n^2 \rangle}$ needs to be replaced by a second-rank tensor. The relative orientation between specimen and illumination is taken into account via tilting the Fresnel propagator. Further details are given in Supplementary Notes 1 and 2.

## Theoretical study

**Forward frozen phonon simulation.** To generate ground truth data for Fig. 2, an approximately $10\,nm \times 10\,nm \times 20\,nm$ large $PbZr_{0.2}Ti_{0.8}O_3$ crystal supercell was constructed. In each unit cell, we randomly picked either a Ti (with 80%) or a Zr atom. Then we sliced the crystal into 50 slices, each with a thickness of a unit cell. We used a standard multislice simulation, with frozen phonons (FP). For each FP configuration, we additionally rolled a Gaussian defocus offset (with a full width at half maximum (FWHM) of 2 nm) and a position offset (with FWHM of 0.05 nm). At each position, the number of thermal configurations was chosen adaptively in the following way. The average of one half of the simulated CBEDs had to have a correlation of at least 0.98 with the other half. Then all CBEDs were averaged. However, we always used at least 50 and at most 150 configurations.

**Reconstruction.** For the atomic position optimization, a model was built in which the Zr/Ti atoms were approximated by virtual atoms, i.e., a linear combination of them. Furthermore, they and the oxygen atoms were set to the symmetry positions within the unit cell, i.e., ferroelectric polarization was set to zero at the start. The model was constructed periodically in beam direction. The optimization then used a batch size of the whole scan. The $L_1$ loss was used, defined for two intensities $I, J$ by

$$\ell(I, J) = \sum_{j,k} |I_{j,k} - J_{j,k}|. \tag{3}$$

After 100 Adam iterations, the $L_1$ loss was almost stable. A learning rate of 1.0 was used, which was halved whenever the loss increased after an update. Using a standard gradient descent (SGD) algorithm will work for most atoms as well. However, the oxygen atoms which reside almost on top of the (ZrTi) site have to be pushed through that column by the algorithm (see Fig. 2e, where the path of the oxygen sites leads through the potential of the Ti/Zr sites). Here, the momentum by Adam helped as shown in Fig. 2, whereas SGD tended to get stuck.

## Experimental study

**4D-STEM acquisition.** Momentum-resolved 4D-STEM data of a $PbZr_{0.2}Ti_{0.8}O_3$ specimen have been collected by scanning a focused aberration-corrected electron probe across the sample and recording a full diffraction pattern at each scan position with an EMPAD detector[31]. The direction of the incident beam was along the [010] zone axis of the crystal. An FEI Titan $G^2$ scanning transmission electron microscope (STEM) equipped with an aberration corrector for the illumination was operated at 200 keV electron energy. The STEM pixel size was set to 35 pm, and a dwell time of 1 ms was chosen for each STEM pixel, synchronized with the frame recording of the EMPAD detector. The probe semi-convergence angle has been measured to 24.6 mrad, and the pixel size in the recorded diffraction patterns was 0.88 mrad.

**Reconstruction.** Initial tilt ($\theta_x = 0.92\,mrad$, $\theta_y = -0.36\,mrad$) and thickness (20 nm)[21], and the $L_1$ loss was minimized (see Supplementary Fig. 4) as in the theoretical study. The whole recorded diffraction pattern was subject to loss calculation up to a detector-imposed maximum scattering angle of 57 mrad. Note that the actual multislice inversion extended much further up to a scattering angle of 110 mrad. Spatial coherence parameters were initialized by sampling a Gaussian with FWHM of one Ångstrom. The defocus offset of the temporal coherence was initialized with zero. Defocus and all other aberrations were initialized to zero, too. Using the recovered phase grating from a single epoch, an atomic model was constructed, again periodic in beam direction. To refine the initial thickness estimate, the loss was calculated for different thicknesses covering a range of $\pm 2.5\,nm$, and the minimum was taken (see Supplementary Fig. 5). The model already captured the ferroelectric polarization, however only qualitatively, as the measurements did not agree with the literature values as summarized in Supplementary Table 1 for comparison. Because the oxygen atom near the (ZrTi) column was invisible, it was placed on the column. Then, the atomic positions were optimized, first for a model without frozen phonons, but with potentials taking the absorptive form factors[17] as well as a Debye-Waller damping into account. This comprises a parametrized single state model for the specimen, which sometimes leads to inaccurate oxygen and (ZrTi) column positions, respectively, as shown in Supplementary Fig. 6. Other parameters, namely probe positions, aberrations, coherence parameters and crystallographic tilt were optimized alongside the atomic positions. These parameters were kept when switching to a full frozen phonon reconstruction, while the atomic positions were reverted to the starting positions. Using Adam and an initial learning rate of 0.05, we optimized for 80 more epochs. All parameters except for the atomic positions were fixed for the first 20 epochs. By this model, which incorporates the exact inversion of the contemporary scattering theory in the presence of TDS and finite coherence, all atomic columns have been reconstructed correctly, as seen in Fig. 4. Also, the loss dropped by approximately 13% as compared to the Debye-Waller based reconstruction scheme with the specimen in a single state.

## Data availability
The raw experimental data generated in this study have been deposited in the Zenodo database[32].

## Code availability
The developed Pytorch program source code is available as Supplementary Software 1.

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

## Acknowledgements

This work was supported by the Bavarian Hightech Agenda (Germany) within the EQAP grant (K.M.-C., Z.H.), the Deutsche Forschungsgemeinschaft under grant number EXC 2089/1 - 390776260 (Germany's Excellence Strategy, K.M.-C.) and the Helmholtz Association (Germany) under contracts VH-NG-1317 (K.M.-C., A.S.), ZT-I-0025 (K.M.-C., B.D., A.S., F.F.) and ZT-I-PF-5-28 (K.M.-C., B.D., F.F.).

## Author contributions

B.D. wrote the Pytorch implementation of forward and inverse multislice, and performed all reconstructions except those of Z.H. Z.H. simulated $PbZr_{0.2}Ti_{0.8}O_3$ and $SrTiO_3$ 4D-STEM data and conducted the pixelwise and parametrized inverse multislice to study the effect of cutoff scattering angle and weight refinement of the pristine atomic columns. A.S. performed the 4D-STEM experiments. B.D., F.F., and K.M.-C. conceived the study, F.F. and K.M.-C. secured the funding. The manuscript was written by B.D. and K. M.-C. All authors discussed the results continuously throughout the project and for final publication.

## Funding

## Competing interests

The authors declare no competing interests.
