## [Peer Review File · Nature Communications]

Exact inversion of partially coherent dynamical electron scattering for picometric structure retrievalReviewer #1 (Remarks to the Author):

The use of the so-called 4D-STEM data set (two dimensions of probe position and two dimensions of diffraction space) is increasingly emerging as a powerful tool for accurate measurements of materials parameters. There are, however, a number of important challenges in inverting such data data. These include the effects of dynamical scattering, partial coherence in the illumination and in the object scattering, and the complexities of the imaging process itself, with phase imaging, for example, having negative regions in the point spread function. The current manuscript describes a method which is based on parameterisation of these effects and describes a method based on state-of-the-art data processing methods to allow the parameters to be found. The approach taken is probably one of the few approaches which can really work, and it is encouraging to see the approach work on real data. Getting the process to work is a tremendous piece of work. I am therefore enthusiastic that the manuscript should be published. The manuscript is generally well-written, though there are additional questions that came to mind during the reading and I do have a range of comments that I would like the authors to address. Starting with the more major comments:

1. I would like to see a little more "under the hood" of the process actually working. I am not clear, for example, what the maximum scattering angle used in the fitting to the experimental data. Is it possible, for example, to extract the simulated diffraction plane intensities for the finally arrived at parameters and to show a comparison with the experiment and perhaps show a difference map.
2. The scattering potential has assumed projected coulomb potentials. Even the potential of independent atoms is not-coulombic, for example due to the screening of the electrons, which is why parameterised atomic scattering factors are used in multislice calculations. What is the impact of reverting to such a simplistic potential model? Presumably the material used has significant ionic character, so were the potentials modified to take account of this, particularly for the oxygen where the effect of charge transfer could be large. What would happen if significant bonding effects led to non-round potentials on the columns?
3. It is implied that the slice thickness in the multislice approach is one whole unit cell, and that it is the projection of the potential through one whole unit cell that is shown in Figure 3. Can this be clarified? If this is the slice thickness, this is larger than is generally used for a multislice. For example, it will not allow for the existence of HOLZ rings which may appear in the experimental data. On a similar vein, 4D-STEM is being used to extract depth dependent information? What are the prospects for doing so in the proposed method?
4. At the bottom of the first column of text on p5, it is correctly pointed out that tilt can affect the measurement of ferroelectric displacements. It is not clear to me in the method how tilt effects are compensated for in the method and I would like this to be clarified.
5. In the methods section, I note that the real-space pixel sampling is larger than Nyquist sampling for a probe with a convergence angle given? Nyquist sample for such a probe is given by $\lambda/(4 \times \text{convergence semi angle})$. This means that the real space sampling of the probe will be undersampling some of the changes that can occur in the detector plane as the probe is scanning. Does the parameterised approach allow for such sub-sampling?

More minor comments:

6. The stoichiometry of the material is incorrectly stated in the abstract.
7. Last line on page 2 should be "due to"
8. Figure 2 panels (e) and (f) have the same label. Perhaps it could be made more explicit that one is referring to the cation position and the other the O anion.
9. I found the paragraph describing Figure 2g to be hard to follow. Can the authors clarify what the implication of different values of negative weight graduate is? Is this essentially a measure of

the error in fit?

10. A minor comment for the first page of the supplementary information: The Wigner approach can reconstruct a strong phase object, which by definition is multiple scattering (ie not single scattering) but neglects propagation. It is therefore good for very strong scattering in a thin sample.

11. The method is making use of both coherent and incoherent scattering, whereas the title suggests only incoherent. Would replacing "incoherent" with "partially coherent" be more in-line with the work in the paper?

Reviewer #2 (Remarks to the Author):

The manuscript presents a novel method to further improve the capabilities of electron ptychography by accounting for thermal diffuse scattering (TDS) in combination with partial coherence in the illuminating probe and therefore obtaining reconstruction results that are in agreement with relativistic scattering theory. The attainable resolution in electron ptychography is currently limited by the thermal fluctuations of the atoms and the authors show how the frozen phonon approach can be integrated into the model of gradient-based reconstruction algorithms to recover the ideal atomic positions as well as the atomic type. One major problem for some of the established reconstruction algorithms is the immense amount of unknowns that arise due to the additional parameters that are required to be recovered alongside the atomic potential when dealing with realistic experimental data. The authors show how the amount of model parameters can be drastically reduced by using a more restrictive parametrization for the potential $V(r)$ and illumination $\psi(r)$. In principle, I believe that the proposed method has some potential to further improve the electron ptychography technique. However, I am not certain to what extent the approach can be applied to more general cases than being demonstrated in the current version of the manuscript and therefore it is questionable whether the novel inversion scheme can outperform the conventional pixelwise based inversion scheme, thus having no advantage over the current state of the art. Before addressing this problem, I will not recommend the publication of this work in Nature Communications. Please see the details below:

1) The authors argue that inversion concepts that follow the conventional pixel-wise strategy violate physics law since they do not include mixed-state recovery of the specimen in their model. I am afraid that this is a false statement. There exist to my knowledge at least one inversion scheme that does allow to retrieve the specimen and the illumination in mixed-states Wakonig, Klaus, et al., Journal of applied crystallography 53.2 (2020): 574-586. Although originally implemented for X-ray ptychography, it has already been used for electron ptychography data Chen, Zhen, et al., Nature communications 11.1 (2020): 2994. and a version for electron ptychography is openly available https://github.com/yijiang1/fold_slice. However, I do agree that ptychography including a mixed-state representation of the object has so far only been presented with experimental data using light Li, Peng, et al., Optics express 24.8 (2016): 9038-9052., but not electrons. I would therefore strongly advice to alter the formulation in the manuscript. Here, it would make sense to highlight the difference of the two approaches. In the conventional approach multiple stationary states $V_i(r)$ are reconstructed while a single set of atom positions, about which the atoms are assumed to wobble, is retrieved in the novel approach.

2) A model for the Coulomb potential $V_{\{FP\}}(r)$ is introduced that includes the projected potential v_Z of an atom that is assumed to be known beforehand. This assumption is not a requirement for the alternative inversion scheme to which the presented method is compared to. The reconstruction of the Coulomb potential $V(r)$ from unknown systems (e.g. high entropy alloys) is, however, an important functionality and therefore the presented inversion scheme seems to be inferior to the pixel-wise reconstruction in terms of generalizability. The authors argue that the inclusion of a weighting factor w_n in their model allow to detect incorrect atomic numbers or missing atoms, but the efficiency of this approach is not presented in the manuscript. A demonstration of the method's performance for unknown systems is yet crucial to fairly compare the two inversion schemes.

3) It is unfortunately hard to grasp the advantage of the introduced inversion scheme over the conventional scheme and thus the real significance of the improvements for electron ptychography. The authors have presented a comparison of the results obtained from the two inversion schemes using data from a $\text{PbZr}_{0.2}\text{Ti}_{0.8}\text{O}_3$ and a PrScO_3 sample. In the first case, a reconstruction result generated with the conventional method is shown in Fig. 1 f and it is obvious that the quality of this reconstruction is inferior to the one shown in Fig. 3 a. However, it is not quite clear how the first reconstruction result was produced; I am almost certain that modal decomposition of the probe has not been applied. In other words, the reconstruction has been generated without making use of the full potential of the pixel-based inversion scheme. The second comparison shown in Supplementary Fig. 2 supports this assumption. In Supplementary Fig. 2 a, where the reconstruction results have been generated from the two inversion schemes with comparable settings, no remarkable difference is visible. In Supplementary Fig. 2 b, where the reconstruction result has been produced with only one probe mode, similar artefacts in the reconstruction are visible as in the reconstruction result shown in Fig. 1 f of the manuscript. I would therefore advice to include a fair comparison between the two inversion schemes, where the improvements made by using the novel inversion scheme are unquestionable.

4) The presented inversion scheme seems like a convenient approach to retrieve the structure of a sample together with the illuminating probe using far less parameters than in the pixelwise based approach. The results shown in the manuscript also indicate its performance on simulated and experimental data. However, my biggest concern relates to the generalizability of the novel inversion scheme. Realistic experimental 4D-STEM data can often be corrupted quite severely for instance from noise, contamination, beam mistilt and much more. The effect on the reconstruction result in the case of the conventional pixel-wise based approach is in form of non-physical artefacts as has been described in the manuscript. Nevertheless, the structure of the sample is often still detectable as has been shown in Fig. 4 of Chen, Zhen, et al., Nature communications 11.1 (2020): 2994., where a very low dose level is used to degrade the data. I am wondering how the proposed method (in particular the generation of the initial atomic model from a preliminary pixel-based reconstruction) would perform in these cases where the data is less optimal. This can be tested straightforwardly by imitating the analysis of the aforementioned work, i.e. applying the novel approach to data with different dose levels.

5) The authors have implemented their method using PyTorch which is the state of the art since it facilitates the calculation of the gradients through automatic differentiation. An implementation in PyTorch of the alternative gradient-based inversion scheme has been proposed for X-ray ptychography in Du, Ming, et al., Optics express 29.7 (2021): 10000-10035. A reference to this work seems to be reasonable and would help to highlight the contribution made by the authors to electron ptychography. Even more important is the open access to the code and the data presented in this manuscript in order to make reproducibility feasible.

6) The authors claim that the pixel-based approach involves more unknowns than necessary and neglects the atomic nature of the specimen. I ask the authors to comment how this statement holds up in light of Van den Broek, Wouter, and Christoph T. Koch., Physical review letters 109.24 (2012): 245502. and Van den Broek, Wouter, and Christoph T. Koch., Physical Review B 87.18 (2013): 184108., where atomicity was explicitly included by assuming a generalized potential by which a sparse potential, comprised mostly of delta-functions at the atom positions would be convolved. While that approach uses a pixel basis, the L1-regularization also constrains the number of possible solutions significantly, by favoring those which result in a few delta-functions.

7) How close does the original guess for the atomic model have to be to the true solution, without the position refinement to fall into a local minimum? I doubt that it is possible to start with a random configuration of atoms and still end up at the correct solution, especially, when working with experimental data.

8) Minor typos:

- Page 2, Fig. 1: 'thermal displacemnts' -> 'thermal displacements'

- Page 3: 'picometre precision' -> 'picometer precision'

- Page 10: 'Supplementary Fig. ??' -> I am not sure to which figure this should refer to.

Reviewer #3 (Remarks to the Author):

I thoroughly enjoyed reading this manuscript and in general I found the work to be highly interesting and novel. The authors have developed an innovative approach to tackle the challenges associated with deciphering atomic structures from ptychographic data obtained using high-energy electron probes focused to subatomic diameters. Unlike traditional pixelwise phase retrieval methods, their parametrized, fully differentiable scheme, inspired by neural network concepts, incorporates essential scattering physics details. This advancement allows for the inversion of ptychographic data using entirely physical quantities. The authors provided compelling evidence supporting the effectiveness of their proposed method, employing both simulations and experimental data. They successfully demonstrated its capability to extract quantitative structural information with picometer-level precision. While the paper is commendable, I have a few points that I would like the authors to consider before publication:

1. From the text it is not entirely clear which method has been used to extract the phase in Fig. 1f. Can it be specified in more detail?
2. Could the authors clarify how the standard deviation of the 2D Gaussian variables describing the shift vector according to the frozen phonon model is included in Eq. (1)?
3. One aspect that needs attention is the specification of the parameters $\{S\}$ and $\{P\}$ that are estimated. On p.5 it is stated that the number of parameters that describe the specimen is usually not more than 100 real parameters which is already far less than the number of atoms contained in an experimental field of view.
4. Along the same line, it is not clear if all the positions of the atoms along the beam direction are estimated or is an 'averaged' result shown? E.g. in Fig. 3b, the caption mentions that it shows a map of reconstructed atomic positions. Does this correspond to a specific slice? Or does it correspond to atomic columns?
5. How are the number of slices in the inversion scheme determined?
6. Could the authors explicitly include the loss function that is used for the optimization and provide more details on the convergence criteria?
7. How sensitive is the algorithm for the choice of the initial parameters?
8. How is this work connected to the work published in *Microscopy and Microanalysis*, 2023, 29, 967–982?
9. Minor suggestions:
 - it would be helpful to be a bit more specific on the type of 'unknowns' in the abstract
 - what are picometer-level atom positions? Do the authors mean e.g. atom positions measured with picometer precision?
 - could the authors clarify/rewrite the following sentence 'Importantly, contemporary inversion schemes reconstruct both multimodal illumination [6–9] and unimodal specimen discretized as pixelated images of complex numbers such that numerical consistency with the experimental observations is achieved.'
 - could the authors indicate $V_{1,2}(r)$ in Fig. 1a?
 - is it possible to show the input potential distribution as well in Fig. 2 (now the reconstructed potential distribution is shown)
 - it was not clear how to interpret the reconstruction pathways in Fig. 2b-f. How can the scale of this figure be linked with Fig. 2a? Does it show the pathways of a column of atoms (as mentioned in the figure or the pathway of an individual atom as mentioned in the text)?
 - could the authors clarify/rewrite the following sentence 'At each position, as many thermal configurations as required for a correlation of > 0.98 of the average of half of the CBEDs with the average of the other half was used, however, at least 50 and at most 150.'
 - in the reconstruction part on page 10 there is a reference to Supplementary Fig. ??
 - could the authors clarify/rewrite the following sentence 'However, the oxygen atoms sitting almost on top of the Ti/Zr site have to be pushed through that column by the algorithm.'

In summary, this manuscript describes some very nice work and the content is excellent.

Response of the authors of

Exact inversion of partially coherent dynamical scattering for picometric structure retrieval

to the reviewer's comments.

We thank the reviewers for the in-depth reading of the paper, and especially for their constructive criticism. In that respect, we appreciate the overall very positive comments a lot, while taking their advices to further improve the manuscript just as seriously. In the following, we provide detailed answers to the points raised. Original comments are shown in black, and our replies in red, whereas modifications in the manuscript are set in bold italic blue font. A manuscript with changes highlighted in red is attached in the file Manuscript_changes.pdf. Changes in the Supplementary are highlighted in Suppl_changes.pdf.

Link to navigate to sections with responses to Reviewer #1, Reviewer #2, Reviewer #3.

Reviewer #1

The use of the so-called 4D-STEM data set (two dimensions of probe position and two dimensions of diffraction space) is increasingly emerging as a powerful tool for accurate measurements of materials parameters. There are, however, a number of important challenges in inverting such data. These include the effects of dynamical scattering, partial coherence in the illumination and in the object scattering, and the complexities of the imaging process itself, with phase imaging, for example, having negative regions in the point spread function. The current manuscript describes a method which is based on parameterisation of these effects and describes a method based on state-of-the-art data processing methods to allow the parameters to be found. The approach taken is probably one of the few approaches which can really work, and it is encouraging to see the approach work on real data. Getting the process to work is a tremendous piece of work. I am therefore enthusiastic that the manuscript should be published. The manuscript is generally well-written, though there are additional questions that came to mind during the reading and I do have a range of comments that I would like the authors to address.

Starting with the more major comments:

1. I would like to see a little more “under the hood” of the process actually working. I am not clear, for example, what the maximum scattering angle used in the fitting to the experimental data. Is it possible, for example, to extract the simulated diffraction plane intensities for the finally arrived at parameters and to show a comparison with the experiment and perhaps show a difference map.

In the Methods section, we now mention further details:

... and the L_1 loss was minimized (see Supplementary Fig. 4) as in the theoretical study. The whole recorded diffraction pattern was subject to loss calculation up to a detector-imposed maximum scattering angle of 57 mrad. Note that the actual multislice inversion extended much further up to a scattering angle of 110 mrad.

Experimental diffraction patterns for selected scan positions as well as their counterparts from the inversion procedure have been added as Supplementary Fig. 3. Since it is impossible to depict this data for the entire scan, we added the experimental and simulated position-averaged

convergent-beam electron diffraction pattern (PACBED) to the figure as well, indicating the trustworthiness of the whole reconstruction.

2. The scattering potential has assumed projected coulomb potentials. Even the potential of independent atoms is not-coulombic, for example due to the screening of the electrons, which is why parameterised atomic scattering factors are used in multislice calculations. What is the impact of reverting to such a simplistic potential model?

In fact, the potentials used in our reconstruction method are the result of Hartree-Fock calculations, which take into account, e.g., screening by electrons in independent atoms exactly. In our case, we use parametrized atomic scattering factors from Lobato et al. [Act. Cryst. A 70, 636 (2014)]. Thus, the same potentials are used for the reconstruction as in contemporary forward multislice simulations. Usually, “Coulomb potential” refers to the potential landscape produced by a set of charges, as present in an atom. This can indeed be more complex than the simple $1/r$ potential created by a point charge to which referee 1 refers to. To avoid any misunderstandings, we now detailed this aspect in conjunction with eq. (1):

Let v_Z be the projected *Coulomb* potential of an atom with atomic number Z , *which is obtained from Hartree-Fock calculations for isolated atoms via contemporary parametrizations* [Act. Cryst. A 70, 636 (2014)]. Assuming ...

Presumably the material used has significant ionic character, so were the potentials modified to take account of this, particularly for the oxygen where the effect of charge transfer could be large. What would happen if significant bonding effects led to non-round potentials on the columns?

The impact of bonding effects and ionicity are indeed aspects to be checked when using parametrized potentials for independent atoms, although modifications to the total potential are extremely low usually. For this revision, we performed additional density functional theory (DFT) simulations of $\text{PbZr}_{0.2}\text{Ti}_{0.8}\text{O}_3$ using the WIEN2K software package. Using the Mott-Bethe relation, Coulomb potentials for this material were obtained as described in previous work cited below. The graphic below summarizes DFT- and isolated atom potential, and includes a line profile across the relevant atomic column. The effects of bonding on the projected potentials are extremely small, and do not lead to a significant anisotropy of the potentials at the atomic sites. Note that the anisotropy for the ZrTi site arises from the projection of the two atoms at slightly different positions and is well-represented by the isolated atom approach. Given that the present approach focuses on position measurement and the integration of TDS whose impact on the diffraction pattern is at least an order of magnitude stronger, we think that including bonding into reconstruction, e.g., via parametrized modified atomic scattering amplitudes (MASA), is a future task that requires substantial effort in a separate project, as suggested at the end of our manuscript.

References the MASA method:

- Müller et al., Phys. Rev. B **81**, 075315 (2010)
- Rosenauer et al., Phys. Rev. B **72**, 085326 (2005)
- Müller-Caspary et al., Ultramicroscopy **178**, 62 (2017)

3. It is implied that the slice thickness in the multislice approach is one whole unit cell, and that it is the projection of the potential through one whole unit cell that is shown in Figure 3. Can

this be clarified? If this is the slice thickness, this is larger than is generally used for a multislice. For example, it will not allow for the existence of HOLZ rings which may appear in the experimental data.

The reconstructed potential is indeed the projection along one unit cell. We would like to point out that this is sufficient to produce the correct HOLZ lines, which we clarify here via several forward multislice simulations (CBED pattern of Strontium Titanate) using different slice thicknesses. For the correct appearance of HOLZ rings, the requirement is that the slicing does not introduce *artificial* periodicities. This condition is definitely met for a slicing strategy that slices at each occurring atomic position, as pointed out correctly by the referee. In general, this

Multislice sim. with different slicings (10nm SrTiO₃, 200kV)

is the most accurate method. In practice, slices are often thicker under the constraint that an integer multiple of the slice thickness Δz equals the lattice periodicity in beam direction, here given by the lattice constant a . Therefore, the upper quadrants with $\Delta z = \frac{a}{2}$ and $\Delta z = a$ yield identical results with HOLZ rings both at the correct positions and negligible differences inside the Bragg discs. In contrast, the bottom part of the figure presents indeed wrong results since $\Delta z = 2a$ halves the periodicity such that additional HOLZ rings appear as artefacts (bottom left). Similarly, $\Delta z = \frac{2}{3}a$ leads to completely wrong results. Note that during the reconstruction, the radius of the experimental HOLZ rings can be used to adjust Fresnel

propagation distances to the correct values.

On a similar vein, 4D-STEM is being used to extract depth dependent information? What are the prospects for doing so in the proposed method?

We have performed simulations to study the actual resolution in electron beam direction. As a general observation, the parametrized approach can deliver z -information with much higher resolution than, e.g., bright- or dark field approaches. We also found that the probability of success to locate an atomic defect in the correct atomic plane depends on the invested electron dose, the specimen thickness, the degree of partial coherence, and the type of defect. For example, a vacant S site could be determined in the correct atomic plane in MoS₂ up to thicknesses of 12 atomic layers with an accuracy of ± 1 layers using at least 2500 e \cdot /Å², whereas a 15 \times higher dose would have been required for HAADF-STEM.

4. At the bottom of the first column of text on p5, it is correctly pointed out that tilt can affect the measurement of ferroelectric displacements. It is not clear to me in the method how tilt effects are compensated for in the method and I would like this to be clarified.

Our approach uses the modification of the Fresnel propagator, which we now firstly state in the Methods section:

The relative orientation between specimen and illumination is taken account via tilting the Fresnel propagator.

Secondly, we added a detailed section on the nature of the tilt and its differentiation in Supplementary Note 2.

5. In the methods section, I note that the real-space pixel sampling is larger than Nyquist sampling for a probe with a convergence angle given? Nyquist sample for such a probe is given by $\lambda/(4 \cdot \text{convergence semi angle})$. This means that the real space sampling of the probe will be undersampling some of the changes that can occur in the detector plane as the probe is scanning. Does the parameterised approach allow for such sub-sampling?

This sampling criterion refers to **direct** ptychography schemes such as single-sideband reconstruction or Wigner distribution deconvolution. These methods build upon Fourier-transforming 4D data with respect to the probe raster. For a semi-angle α_S , information is transferred up to the frequency $2\alpha_S/\lambda$, whereas twice this value is usually taken to allow a suitable overlap of the so-called “trotters”. For **iterative** schemes, as used here and in the (e)PIE approach, the criterion mentioned by referee 1 does not apply (no Fourier transform is involved with respect to the scan). In fact, these schemes tend to utilize highly convergent but defocused probes with rather coarse scan rasters to obtain structures with significant super-resolution.

More minor comments:

6. The stoichiometry of the material is incorrectly stated in the abstract. **Done.**

7. Last line on page 2 should be “due to”. **Done.**

8. Figure 2 panels (e) and (f) have the same label. Perhaps it could be made more explicit that one is referring to the cation position and the other the O anion.

Done, site labelled additionally.

9. I found the paragraph describing Figure 2g to be hard to follow. Can the authors clarify what the implication of different values of negative weight graduate is? Is this essentially a measure of the error in fit?

We extended the paragraph to yield a comprehensive explanation of the Fig. 2g:

*The greatly increased sensitivity of our inversion method to the chemical composition via Z-contrast, arising from the exact inclusion of high-angle TDS, is shown in Fig. 2g. 4D-STEM data have been simulated for a random alloy (Zr_xTi_{1-x}) as before, except that the composition x fluctuated statistically, as depicted for four selected columns with Zr fractions x between 14 and 26%. In practice, only the average Zr fraction of 20% may be known which we used to initialise the parametric model $\{S\}$ with a virtual ($Zr_{0.2}Ti_{0.8}$) atom. **Its potential is approximated by the linear combination of 20% the potential of Zr and 80% Ti, corresponding to an effective atomic number of $Z_{\text{eff}}=25.6$.***

*None of the mixed columns I-IV used to simulate the diffraction patterns adopts this composition exactly. Consequently, the inversion procedure captures this discrepancy by detecting too strong scattering potentials, or equivalently, too high weights w_n in eq. (1) in the model at (ZrTi) sites I and II, and vice versa at sites III and IV. For sites I-IV, the negative gradient of the loss with respect to the weights was determined using an inverse model that includes TDS via frozen phonons (blue), or neglects thermal disorder within the Debye-Waller approach (red), respectively. In both cases, a clear linear relation can be seen, **whereas only the FP gradient nearly vanishes in case the model adopts already the true composition (black dot in Fig. 2g).** Neglecting TDS (red) in the inverse multislice scheme would thus lead to the wrong chemical composition of the ($Zr_{0.2}Ti_{0.8}$) column. This study not only demonstrates that*

a misestimated chemistry in an initial parametrization of {S} in mixed columns can be corrected, it also confirms that treating (ZrTi) as a virtual atom is justified here which will be utilized in the experimental study below.

10. A minor comment for the first page of the supplementary information: The Wigner approach can reconstruct a strong phase object, which by definition is multiple scattering (ie not single scattering) but neglects propagation. It is therefore good for very strong scattering in a thin sample.

The derivation of the Wigner approach starts by expressing the specimen exit wave as a product of the wave function of the illuminating probe with a complex object transmission function. In this sense, the approach is based on a single-interaction scheme. We agree that the Wigner approach can handle strong phase objects, and is good for strongly scattering thin specimens. Nevertheless, we would like to point out that this does not tackle effects of multiple scattering, at least in the form we are aware of. To support our statement, we supply a concise but complete mathematical derivation in an extra document **Wigner_derivation_referee_1.pdf** for reviewer #1. We are ready to modify our statement in the Supplementary, or to take it out completely, should there be a mistake in our understanding or debate arising, since the Wigner method is not central to the results of our paper dealing with inverse multislice.

11. The method is making use of both coherent and incoherent scattering, whereas the title suggests only incoherent. Would replacing “incoherent” with “partially coherent” be more in-line with the work in the paper?

Indeed, illumination is partially coherent and TDS fully incoherent. We changed the title to partially coherent, which also includes the incoherent case as one extreme and state in the text how ensemble averages are taken where appropriate.

Reviewer #2

The manuscript presents a novel method to further improve the capabilities of electron ptychography by accounting for thermal diffuse scattering (TDS) in combination with partial coherence in the illuminating probe and therefore obtaining reconstruction results that are in agreement with relativistic scattering theory. The attainable resolution in electron ptychography is currently limited by the thermal fluctuations of the atoms and the authors show how the frozen phonon approach can be integrated into the model of gradient-based reconstruction algorithms to recover the ideal atomic positions as well as the atomic type. One major problem for some of the established reconstruction algorithms is the immense amount of unknowns that arise due to the additional parameters that are required to be recovered alongside the atomic potential when dealing with realistic experimental data. The authors show how the amount of model parameters can be drastically reduced by using a more restrictive parametrization for the potential $V(r)$ and illumination $\psi(r)$. In principle, I believe that the proposed method has some potential to further improve the electron ptychography technique. However, I am not certain to what extent the approach can be applied to more general cases than being demonstrated in the current version of the manuscript and therefore it is questionable whether the novel inversion scheme can outperform the conventional pixelwise based inversion scheme, thus having no advantage over the current state of the art. Before addressing this problem, I will not recommend the publication of this work in Nature Communications. Please see the details below:

1) The authors argue that inversion concepts that follow the conventional pixel-wise strategy violate physics law since they do not include mixed-state recovery of the specimen in their model. I am afraid that this is a false statement. There exist to my knowledge at least one inversion scheme that does allow to retrieve the specimen and the illumination in mixed-states Wakonig, Klaus, et al., Journal of applied crystallography 53.2 (2020): 574-586. Although originally implemented for X-ray ptychography, it has already been used for electron ptychography data Chen, Zhen, et al., Nature communications 11.1 (2020): 2994. and a version for electron ptychography is openly available https://github.com/yijiang1/fold_slice. However, I do agree that ptychography including a mixed-state representation of the object has so far only been presented with experimental data using light Li, Peng, et al., Optics express 24.8 (2016): 9038-9052., but not electrons. I would therefore strongly advice to alter the formulation in the manuscript.

Concerning the statements on contemporary mixed-state methodologies, we checked and overworked the manuscript with great care, addressing the references given by referee #2 and beyond.

a) The whole manuscript deals with 4D data recorded in contemporary electron microscopes. To distinguish from X-ray methodologies, we changed the title to refer to *partially coherent electron scattering*.

b) The reference Wakonig, *J.Appl.Cryst.* 53 (2020) (added now) reports on the software PtychoShelves, in which the reference to the already cited Thibault and Menzel (2013) work on the physical model is given in the context of mixed-state modes. We thank the referee for pointing out that Thibault and Menzel refer to mixed-state objects already, and corrected the citation context. Importantly, applications such as Li, *Optics Express* 24, 9038 (2016) (ref. added) employed light-optical experiments and assumed single scattering.

c) The reference *Chen, Zhen, et al., Nature communications 11.1 (2020)* had already been included in the first submission, in a context where we refer to ptychography reconstructions that indeed allow for considering mixed-state illumination. In this respect, referee #2 is right that Chen2020 performed a mixed-state reconstruction, but to our best knowledge after reading the paper carefully, *not* with respect to the *specimen*.

The text was modified to:

While multimodal specimens have been proposed [Thibault2013, Wakonig2020] and applied to optical experiments neglecting multiple scattering [Li2016], working out the conceptual framework for partially coherent inverse multiple electron scattering and the application to experimental data remains outstanding.

Here, it would make sense to highlight the difference of the two approaches. In the conventional approach multiple stationary states $V_i(\mathbf{r})$ are reconstructed while a single set of atom positions, about which the atoms are assumed to wobble, is retrieved in the novel approach.

In the introduction, we now highlight the difference to our approach. We note here that a pixelwise reconstruction (phase) *itself* does usually not form the final scientific conclusion; the atomically resolved phase gratings are nearly always interpreted in terms of the potentially underlying atomic structure concerning positions, types, defects anyway. Due to the nonlinearity of the pixelwise inversion, the regularizations, and the nonlinear fitting of the atomic model to the obtained phase grating (instead of directly to the experimental data, as we do), pixelwise models sometimes contain more steps than necessary. We added:

Additionally, the pixelwise inversion strategy is not confined to yield physically plausible Coulomb potentials $V(\vec{r})$ and wavefunctions $\psi(\vec{r})$. Furthermore, any atomistic interpretation has to be inferred a posteriori, by means of image processing being independent of the experimental data.

2) A model for the Coulomb potential $V_{\{FP\}}(\mathbf{r})$ is introduced that includes the projected potential v_Z of an atom that is assumed to be known beforehand. This assumption is not a requirement for the alternative inversion scheme to which the presented method is compared to. The reconstruction of the Coulomb potential $V(\mathbf{r})$ from unknown systems (e.g. high entropy alloys) is, however, an important functionality and therefore the presented inversion scheme seems to be inferior to the pixel-wise reconstruction in terms of generalizability.

The dichotomy mentioned by referee #2 is not claimed in the manuscript, and we are indeed convinced that both parametrized and pixelwise schemes have individual and complementary merits in different situations. In fact, we guess our initial model based on the pixelwise scheme. Note that one of our main intentions was to develop an inverse approach taking into account thermal diffuse scattering, for which we need the atomic parametrization. We agree with referee #2 that this strategy requires a suitably good initial guess for the atomic structure, which is reliably possible for the PZT ferroelectric we dealt with, but might be more difficult to set up for high entropy alloys or even organic matter. Still there, after obtaining an atomically resolved pixelwise phase grating, subsequent transition to the parametrized exact inversion scheme presented here, and harvesting the Z-sensitivity from thermal diffuse scattering at high angles is a powerful future direction to broaden the range of application. In this sense, the atomically parametrized scheme presented in this work is the most generic and generalizable at present since scattering physics is comprehensively incorporated in the inverse model. Please also see the study on atomic types at point 7.

The authors argue that the inclusion of a weighting factor w_n in their model allow to detect incorrect atomic numbers or missing atoms, but the efficiency of this approach is not presented in the manuscript. A demonstration of the method's performance for unknown systems is yet crucial to fairly compare the two inversion schemes.

We agree that this aspect had indeed been under-represented in our first submission. We have added a study in which we demonstrate that optimizing the weights of the Pb and O sites with a wrongly guessed initial model leads to the correct stoichiometry of the structure, which completes the analysis of the mixed atomic column in Fig. 2g. The results are presented in the newly added Fig. 3

We close the theoretical study by investigating the capabilities of the parametric approach to detect the correct atomic types, since they might not be known exactly in practice. To this end, we optimised the weights w_n in eq. (1) for initial models in which we substituted either lead or oxygen by the atomic species shown in Fig. 3b,c. Indeed, the graphs show that $w_n = 1$ is obtained for the correct occupation with O and Pb according to the $PbZr_{0.2}Ti_{0.8}O_3$ ground truth. Importantly, erroneously starting with a PbS or PbSe host crystal as in Fig. 3b suggests to weigh the S and Se potential by approximately 50 and 35%, respectively. Note that also the Pb weights have been optimised and remain close to one. Similarly, a fictitious substitution of Pb by Hg, Tl, Bi or Po shown in Fig. 3c yields an almost linearly adapted increase or decrease of the scattering potential for too light or too heavy atoms, respectively. While substitution at the Pb site changes also the oxygen weights slightly by less than 5%, a confusion with S or Se can be ruled out according to Fig. 3b.

Fig. 3: Detection of correct atom types. Simulation of 4D-STEM data for the model in a has been performed using frozen phonons. The initial model has been set up by placing b oxygen and c lead sites with different atom types. Shown are the optimized weights after 20 epochs employing an inverse model with 10 FP configurations. The weights clearly indicate O and Pb as the correct solution, and suggest lowering or increasing the atomic number for too heavy or light atoms, respectively.

3) It is unfortunately hard to grasp the advantage of the introduced inversion scheme over the conventional scheme and thus the real significance of the improvements for electron ptychography.

We summarize the arguments outlined above.

- (1) It captures the scattering physics in exact manner in the omnipresence of TDS in thick specimens.
- (2) It has significantly fewer free parameters, making the reconstruction more stable and less dependent on ad-hoc choices like regularization parameters.

- (3) As parameters of interest are directly encoded and measured from the 4D experiment, no postprocessing is necessary. Hence, the image processing part of the data processing pipeline is merged into the reconstruction step and thus depends on the experimental data.
- (4) Only physical solutions for the probe wave function and the specimen potential are allowed for.
- (5) Artefact-free highly precise atomic positions and polarisation in a ferroelectric is presented in simulation and comprehensive application.

The authors have presented a comparison of the results obtained from the two inversion schemes using data from a $\text{PbZr}_{0.2}\text{Ti}_{0.8}\text{O}_3$ and a PrScO_3 sample. In the first case, a reconstruction result generated with the conventional method is shown in Fig. 1 f and it is obvious that the quality of this reconstruction is inferior to the one shown in Fig. 3 a. However, it is not quite clear how the first reconstruction result was produced; I am almost certain that modal decomposition of the probe has not been applied. In other words, the reconstruction has been generated without making use of the full potential of the pixel-based inversion scheme. The second comparison shown in Supplementary Fig. 2 supports this assumption. In Supplementary Fig. 2 a, where the reconstruction results have been generated from the two inversion schemes with comparable settings, no remarkable difference is visible. In Supplementary Fig. 2 b, where the reconstruction result has been produced with only one probe mode, similar artefacts in the reconstruction are visible as in the reconstruction result shown in Fig. 1 f of the manuscript. I would therefore advice to include a fair comparison between the two inversion schemes, where the improvements made by using the novel inversion scheme are unquestionable.

We clarified that in Fig. 1f we indeed used a multistate probe, relying on our direct parametrization of partial spatial coherence. As the referee points out correctly, this parametrization yields a visually similar result to the unimodal approach (Supplementary Note 3, part (b) of the graphic). The former Fig. 1f has, however, been generated without any regularization during the reconstruction. Our intention had originally been to not introduce further degrees of freedom (regularization parameters) for the reconstruction, and to demonstrate that all diffraction features (Fig. 1g) can be reproduced rather accurately up to 60 mrad, incl. TDS. From the perspective of a fair comparison of both strategies (pixelwise and parametrized) we definitely agree with referee #2 that a pixelwise result obtained under more convenient settings needs to be added, which we did by tiling Figs. 1f,g such that the regularized and non-regularized result are both shown. From Fig. 1g, one sees that the regularization tends to suppress the TDS particularly in diffraction space in favour of smoother phase gratings in real space. Please note that the acquisition conditions were closer to focus as usual for pixelwise schemes, because of the domain structure of the ferroelectric and the experienced better performance of the parametrized approach. In conclusion, Fig. 1f is now showing a definitely improved pixelwise result, and we modified the text to:

While partial coherence of the illumination was included, a unimodal specimen was reconstructed without (bottom left) and with regularization to enforce $\varphi(\vec{r})$ to be continuous (top right). [...] Although it is physically impossible to produce TDS without any ensemble averaging, the non-regularized result in Fig. 1g exhibits a nearly 1:1 agreement with the experiment in Fig. 1d. [...] A detailed simulation study is shown in Supplementary Fig. 1. On the other hand, regularization strategies can assure smooth and continuous phases top right in Fig. 1f, but are incapable of producing the details of thermal diffuse scattering in diffraction space as seen from the corresponding PACBED top right in Fig. 1g. Though the quality of pixelwise reconstructed phases depends on acquisition settings and regularization

details, explicitly accounting for phonons in the solution to the inverse problem is crucial to exploit the entire information in 4D-STEM experiments to include especially high-angle scattering.

4) The presented inversion scheme seems like a convenient approach to retrieve the structure of a sample together with the illuminating probe using far less parameters than in the pixelwise based approach. The results shown in the manuscript also indicate its performance on simulated and experimental data. However, my biggest concern relates to the generalizability of the novel inversion scheme. Realistic experimental 4D-STEM data can often be corrupted quite severely for instance from noise, contamination, beam mistilt and much more. The effect on the reconstruction result in the case of the conventional pixel-wise based approach is in form of non-physical artefacts as has been described in the manuscript. Nevertheless, the structure of the sample is often still detectable as has been shown in Fig. 4 of Chen, Zhen, et al., Nature communications 11.1 (2020): 2994., where a very low dose level is used to degrade the data. I am wondering how the proposed method (in particular the generation of the initial atomic model from a preliminary pixel-based reconstruction) would perform in these cases where the data is less optimal. This can be tested straightforwardly by imitating the analysis of the aforementioned work, i.e. applying the novel approach to data with different dose levels.

We thank referee #2 for his constructive comments here. Indeed, beam mistilt is already accounted for in our approach, and the dose setting for our PZT specimen could be set high enough (material stability) so as to work in a sufficiently high SNR regime. And we fully agree that the underlying structure is often still detectable as in Chen2020. We think that our present work has demonstrated the benefits of atom position detection precision for PZT, and decided to outline a dose study for a material where the dose is indeed a critical parameter, MoS₂ monolayers. To this end, we simulated an MoS₂ monolayer 4D STEM scan with partial spatial and partial temporal coherence effects, and in frozen phonon mode. The atomic structure was intact except for a single sulfur-vacancy. The question that we then addressed was: Given a certain dose, how large is the probability to successfully detect the S-vacancy? To this end, we used a fully intact initial model without vacancy and determined the gradients of the atomic weights from the loss derived from the simulated 4D data. For each dose level, Poissonian noise was rolled many times, and we considered the vacancy detection as a success if the weight gradient at the known position of the missing S atom was maximum. For a comparison, this was done using the full 4D STEM data, a bright and a dark field detector (BF, HAADF).

We think that this result below demonstrates the benefits of parametrization at the atomic level clearly, since gradients are known directly for the atom types involved, and render highly dose-efficient as compared to the BF and HAADF counterparts. It also encourages a wide field of future applications of parametrized 4D-STEM ptychography in low-dose applications where specific details at the atomic level need to be deciphered.

5) The authors have implemented their method using PyTorch which is the state of the art since it facilitates the calculation of the gradients through automatic differentiation. An implementation in PyTorch of the alternative gradient-based inversion scheme has been proposed for X-ray ptychography in Du, Ming, et al., Optics express 29.7 (2021): 10000-10035. A reference to this work seems to be reasonable and would help to highlight the contribution made by the authors to electron ptychography. Even more important is the open access to the code and the data presented in this manuscript in order to make reproducibility feasible.

We fully agree and cited the given reference. Open access to code and data will be accomplished together with the publication and is opted for during submission. We included a zipped version of our code and an example program in our resubmission.

6) The authors claim that the pixel-based approach involves more unknowns than necessary and neglects the atomic nature of the specimen. I ask the authors to comment how this statement holds up in light of Van den Broek, Wouter, and Christoph T. Koch., Physical review letters 109.24 (2012): 245502. and Van den Broek, Wouter, and Christoph T. Koch., Physical Review B 87.18 (2013): 184108., where atomicity was explicitly included by assuming a generalized potential by which a sparse potential, comprised mostly of delta-functions at the atom positions would be convolved. While that approach uses a pixel basis, the L1-regularization also constrains the number of possible solutions significantly, by favoring those which result in a few delta-functions.

The two mentioned papers by Van den Broek et al indeed have an alternative parametrization for atomic potentials. However, as the referee correctly points out, the number of parameters is not reduced. Furthermore, different atomic types are approximated by rescaling a single prototype. Also, it is unclear how subpixel positions can be retrieved from a vector that is only approximately a sum of a few delta peaks. Neither does it cover a multi-state specimen. This approach is not documented on an experimental ptychographic dataset in these papers. We cannot comment on whether it could be extended and used in our setting, which is the reason we refrain from speculating in that direction.

Without any doubt, the papers by Van den Broek et al are extremely important, as they laid the ground for complex models in ptychography and beyond, that are optimized via an automatic differentiation scheme. They were an important inspiration for us and are thus already cited prominently in our paper.

7) How close does the original guess for the atomic model have to be to the true solution, without the position refinement to fall into a local minimum? I doubt that it is possible to start with a random configuration of atoms and still end up at the correct solution, especially, when working with experimental data.

The algorithm is based in a suitable guess of the atomic positions for the initial model. The exact conditions may be case dependent, but as referee #2 points out (4.), a structural guess can usually be obtained from a pixelwise reconstruction. In our PZT simulation study in Fig 2, symmetry positions were enough to perfectly detect the polarization. In our experimental study, we placed O_3 exactly on the heavy Ti/Zr column, making it deliberately hard for our algorithm. In our provided software, we give a simple simulated example, where distorting the positions of STO randomly by up to 10% of a unit cell could easily be recovered.

Importantly, we added a comprehensive study on the correct detection of atomic types as Fig. 3 in the resubmission, as detailed in conjunction with the weight optimization at point 3 of referee #2.

8) Minor typos:

- Page 2, Fig. 1: 'thermal displacemnts' -> 'thermal displacements'. **Done.**
- Page 3: 'picometre precision' -> 'picometer precision'. **Done.**
- Page 10: 'Supplementary Fig. ??' -> I am not sure to which figure this should refer to.

We removed the reference.

Reviewer #3

I thoroughly enjoyed reading this manuscript and in general I found the work to be highly interesting and novel. The authors have developed an innovative approach to tackle the challenges associated with deciphering atomic structures from ptychographic data obtained using high-energy electron probes focused to subatomic diameters. Unlike traditional pixelwise phase retrieval methods, their parametrized, fully differentiable scheme, inspired by neural network concepts, incorporates essential scattering physics details. This advancement allows for the inversion of ptychographic data using entirely physical quantities. The authors provided compelling evidence supporting the effectiveness of their proposed method, employing both simulations and experimental data. They successfully demonstrated its capability to extract quantitative structural information with picometer-level precision. While the paper is commendable, I have a few points that I would like the authors to consider before publication:

1. From the text it is not entirely clear which method has been used to extract the phase in Fig. 1f. Can it be specified in more detail?

Several additions were made to Figs. 1f,g and the related text: We added (see also point 3 of referee #2):

While partial coherence of the illumination was included, a unimodal specimen was reconstructed without (bottom left) and with regularization to enforce $\varphi(\vec{r})$ to be continuous (top right). [...] Although it is physically impossible to produce TDS without any ensemble averaging, the non-regularized result in Fig. 1g exhibits a nearly 1:1 agreement with the experiment in Fig. 1d. [...] A detailed simulation study is shown in Supplementary Fig. 1. On the other hand, regularization strategies can assure smooth and continuous phases top right in Fig. 1f, but are incapable of producing the details of thermal diffuse scattering in diffraction space as seen from the corresponding PACBED top right in Fig. 1g. Though the quality of pixelwise reconstructed phases depends on acquisition settings and regularization details, explicitly accounting for phonons in the solution to the inverse problem is crucial to exploit the entire information in 4D-STEM experiments to include especially high-angle scattering.

2. Could the authors clarify how the standard deviation of the 2D Gaussian variables describing the shift vector according to the frozen phonon model is included in Eq. (1)?

We thank referee #3 for pointing out ambiguities in the terminology. We now

- introduce the vector describing the thermal displacements \vec{u} where initially occurring in conjunction with Fig. 1a at the beginning of the Results section,

- corrected the description of $\vec{g}_{n,\tau}$ in conjunction with eq. (1) to “*identically distributed two dimensional normal Gaussian variables*” and explicitly refer to $\langle \vec{u}^2 \rangle$ as the *mean squared thermal displacement*,

- detailed the explicit calculation in the Methods section to: *Eq. (1) is for clarity written for isotropic thermal vibrations. For each atomic site, these are characterised by the mean squared thermal displacements $\langle \vec{u}^2 \rangle$ along one dimension, relating to the isotropic Debye parameter via $B = 8\pi^2 \langle \vec{u}^2 \rangle$ and to the Debye-Waller factor as $\exp(-\frac{1}{4}Bq^2)$ with spatial frequency q . For anisotropic cases, $\sqrt{\langle \vec{u}^2 \rangle}$ needs to be replaced by a second-rank tensor.*

3. One aspect that needs attention is the specification of the parameters {S} and {P} that are estimated. On p.5 it is stated that the number of parameters that describe the specimen is usually not more than 100 real parameters which is already far less than the number of atoms contained in an experimental field of view.

Indeed, we mention “a few hundred” at the beginning of the manuscript which is true, and corrected the second statement accordingly from “*not more than 10²*” to “*a few hundred*”.

4. Along the same line, it is not clear if all the positions of the atoms along the beam direction are estimated or is an ‘averaged’ result shown? E.g. in Fig. 3b, the caption mentions that it shows a map of reconstructed atomic positions. Does this correspond to a specific slice? Or does it correspond to atomic columns?

The symmetry of the specimen and correcting a mistilt from zone axis [010] by a (reconstructed) tilt of the Fresnel propagator justifies using periodicity along beam direction (except for thermal displacements), as stated in the Methods section. Correspondingly, Fig. 4b (former 3b) shows atomic *column* positions, as now stated explicitly in the caption.

5. How are the number of slices in the inversion scheme determined?

We explain that now in the method section:

To refine the initial thickness estimate, the loss was calculated for different thicknesses covering a range of ± 2.5 nm and the minimum was taken (see Supplementary Fig. ~5).

Supplementary Fig. 5 showing loss vs. thickness was added.

6. Could the authors explicitly include the loss function that is used for the optimization and provide more details on the convergence criteria?

We used the L_1 loss, which we now give explicitly in the method section:

The L_1 was used, defined for two intensities I, J by

$$\ell(I, J) = \sum_{j,k} |I_{j,k} - J_{j,k}| .$$

Other popular choices like the Poisson log-likelihood or the amplitude loss have to be treated more carefully, due to a singularity the loss function or in the derivative, respectively. We found that in our rather high dose regime, L_1 loss suffices.

Regarding the convergence, we waited until the loss stagnated, then halved the learning rate. If the stagnation prevailed, we either switched to a more sophisticated model (frozen phonon) or stopped. We added Supplementary Fig. 4 showing the loss to justify this.

7. How sensitive is the algorithm for the choice of the initial parameters?

[see also referee #2, 7.] The algorithm is based in a suitable guess of the atomic positions for the initial model. The exact conditions may be case dependent, but as also referee #2 points out, a structural guess can usually be obtained from a pixelwise reconstruction. In our PZT simulation study in Fig 2, symmetry positions were enough to perfectly detect the polarization.

In our experimental study, we placed O_3 exactly on the heavy Ti/Zr column, making it deliberately hard for our algorithm. In our provided software, we give a simple simulated example, where distorting the positions of STO randomly by up to 10% of a unit cell could easily be recovered.

Importantly, we added a comprehensive study on the correct detection of atomic types as Fig. 3 in the resubmission and write:

We close the theoretical study by investigating the capabilities of the parametric approach to detect the correct atomic types, since they might not be known exactly in practice. To this end, we optimised the weights w_n in eq. (1) for initial models in which we substituted either lead or oxygen by the atomic species shown in Fig. 3b,c. Indeed, the graphs show that $w_n = 1$ is obtained for the correct occupation with O and Pb according to the $PbZr_{0.2}Ti_{0.8}O_3$ ground truth. Importantly, erroneously starting with a PbS or PbSe host crystal as in Fig. 3b suggests to weigh the S and Se potential by approximately 50 and 35%, respectively. Note that also the Pb weights have been optimised and remain close to one. Similarly, a fictitious substitution of Pb by Hg, Tl, Bi or Po shown in Fig. 3c yields an almost linearly adapted increase or decrease of the scattering potential for too light or too heavy atoms, respectively. While substitution at the Pb site changes also the oxygen weights slightly by less than 5%, a confusion with S or Se can be ruled out according to Fig. 3b.

8. How is this work connected to the work published in Microscopy and Microanalysis, 2023, 29, 967–982?

There are fundamental differences as to concept and application. First, the scattering matrices recover structure factors for ideal crystals, with temperature taken into account in Debye-Waller approximation. Including TDS would inflate the scattering matrix to an extent that is numerically not manageable, given that accounting for N Fourier coefficients causes the S-matrix to be $N \times N$, and here thermal averages are not even included. Let a multislice support in reciprocal space be of the order 300×300 pixels, this would cause the S-matrix to be $(300 \times 300)^2 \approx 90,000 \times 90,000$. On top of that, Sadri et al. present a sole simulation-based study without taking partial coherence into account at all. We thank the reviewer for pointing out this reference, and added it now in the discussion:

Simultaneously, incoherent thermal diffuse scattering so far inaccessible by scattering matrix [Sadri2023] or inverse multislice methods is incorporated in exact manner,...

9. Minor suggestions:

- it would be helpful to be a bit more specific on the type of ‘unknowns’ in the abstract

We rewrote this to clarify that each pixel is a free parameter:

By pixelwise phase retrieval, current approaches do not only involve orders of magnitude more free parameters than necessary, [...].

- what are picometer-level atom positions? Do the authors mean e.g. atom positions measured with picometer precision?

Indeed, we reformulated to [...] are measured with picometer precision in the abstract.

- could the authors clarify/rewrite the following sentence ‘Importantly, contemporary inversion schemes reconstruct both multimodal illumination [6–9] and unimodal specimen discretized as pixelated images of complex numbers such that numerical consistency with the experimental observations is achieved.’

We agree that this sentence is too convoluted. We meant that the specimen potential and the illumination wave (or multiple waves in case of a multimodal illumination) are discretized on a fine grid, forming a complex-valued image. We changed the text as follows.

Importantly, contemporary inversion schemes reconstruct both multimodal illumination and unimodal specimen. Both, the potential of the specimen as well as the states of the illumination, are discretized on a fine grid, as pixelated images. These complex-valued images are then optimized to achieve numerical consistency with the experimental observations.

- could the authors indicate $V_{1,2}(r)$ in Fig. 1a?

This was done and is now also explained in the caption.

- is it possible to show the input potential distribution as well in Fig. 2 (now the reconstructed potential distribution is shown)

We added Supplementary Fig. 2 and referred to it in the text.

- it was not clear how to interpret the reconstruction pathways in Fig. 2b-f. How can the scale of this figure be linked with Fig. 2a? Does it show the pathways of a column of atoms (as mentioned in the figure or the pathway of an individual atom as mentioned in the text)?

We clarified in the caption: *The rectangular patches b-f represent magnified regions in a with the respective colour.*

In accordance with point 4 of referee #3, we brought Fig. 2 and the respective paragraph in the text to consistency by now referring to *atom columns* explicitly where appropriate.

- could the authors clarify/rewrite the following sentence ‘At each position, as many thermal configurations as required for a correlation of > 0.98 of the average of half of the CBEDs with the average of the other half was used, however, at least 50 and at most 150.’

Another convoluted sentence, which we reformulated, and which is hopefully clearer now.

At each position, the number of thermal configurations was chosen adaptively in the following way. The average of one half of the simulated CBEDs had to have a correlation of at least 0.98 with the other half. Then all CBEDs were averaged. However, we always used at least 50 and at most 150 configurations.

- in the reconstruction part on page 10 there is a reference to Supplementary Fig. ??

See comment of referee #2. We removed the reference.

- could the authors clarify/rewrite the following sentence ‘However, the oxygen atoms sitting almost on top of the Ti/Zr site have to be pushed through that column by the algorithm.’

We now refer to Figure 2. e, where one can see how the path of the oxygen positions has to pass through the potential of the Ti/Zr atoms:

(see Fig. ~2e, where the path of the oxygen sites leads through the potential of the Ti/Zr sites).

In summary, this manuscript describes some very nice work and the content is excellent.

Reviewer #1 (Remarks to the Author):

I really do appreciate the authors' careful consideration of my comments which has been a very interesting discussion. Overall, I am very happy with the responses to my comments and the associated changes. I just have a few very minor further queries. I do not need to see the responses to these and I am happy for the paper to be accepted after the authors have considered them.

I may have missed it somewhere, but what phase retrieval method was used to generate the image in Figure 1f. In particular, was the multislice Ptychography method used? Can the simulated and experimental sides of the figure be labelled?

Similarly, I don't think I spotted whether the method built on an existing multislice simulation code for the forward simulations. If so, can it be stated which one?

In the authors' response to my point 5, I would point out that the sampling for ePIE is often expressed in terms of a percentage overlap of probes, but for a focused probe the overlap is essentially equivalent to the Nyquist sampling for the probe despite a Fourier transform with respect to probe position not being used. My question was really whether the authors' approach allowed for further relaxation of the sampling by having less probe overlap than needed by ePIE. If so, then this is a useful advantage of the authors' method.

I agree that the discussion around my point 10 is some esoteric and not central to the work presented. It can be challenging to related discrete sampling events to a wave model, but my understanding is that the weak phase approximation (ie the truncation to the first term of the power series expansion of the phase) is regarded as a single scattering approximation (my recollection is that there is early work by Cowley on this). Higher order terms (ie the strong phase approximation) contain multiple scattering events, but in a thin sample. The term dynamical diffraction refers to multiple scattering with propagation (as included in a multislice calculation. I accept that the concept of multiple scattering in a thin sample (ie a uranium atom) is somewhat abstract. Perhaps the way this complication can be avoided is to simply assert that Wigner requires a multiplicative transmission function, and leave it at that.

Reviewer #2 (Remarks to the Author):

The revised manuscript "Exact inversion of partially coherent dynamical electron scattering for picometric structure retrieval" by Diederichs et al. has been significantly improved and has addressed all my concerns. It is now much more clear how the proposed approach compares to the conventional pixelwise based inversion scheme. I believe that the proposed method has the potential to further improve the electron Ptychography technique and I am happy to recommend the manuscript to be published in Nature Communications now.

Reviewer #3 (Remarks to the Author):

I have carefully reviewed the revised manuscript and I am impressed with the thoroughness of the revisions made by the authors. The authors have developed an innovative approach to tackle the challenges associated with deciphering atomic structures from Ptychographic data obtained using high-energy electron probes focused to subatomic diameters. Unlike traditional pixelwise phase retrieval methods, their parametrized, fully differentiable scheme, inspired by neural network concepts, incorporates essential scattering physics details. This advancement allows for the inversion of Ptychographic data using entirely physical quantities. The authors provided compelling evidence supporting the effectiveness of their proposed method, employing both simulations and experimental data. They successfully demonstrated its capability to extract quantitative structural information with picometer-level precision. The changes made in the revised version significantly strengthened the paper, addressing previous concerns and improving the overall clarity and coherence of the content. I commend the authors for their meticulous attention to detail and their

efforts to enhance the quality of the manuscript. Based on the revisions made and the high quality of the work presented, I strongly recommend that the paper be accepted for publication in its current form.

Reply to final comments Reviewer 1:

I really do appreciate the authors' careful consideration of my comments which has been a very interesting discussion. Overall, I am very happy with the responses to my comments and the associated changes. I just have a few very minor further queries. I do not need to see the responses to these and I am happy for the paper to be accepted after the authors have considered them.

I may have missed it somewhere, but what phase retrieval method was used to generate the image in Figure 1f. In particular, was the multislice ptychography method used? Can the simulated and experimental sides of the figure be labelled?

Labels were added to Fig. 1f (with and without regularization). We added to the corresponding manuscript text: [...] by the pixelwise inverse multislice result [...]

Similarly, I don't think I spotted whether the method built on an existing multislice simulation code for the forward simulations. If so, can it be stated which one?

The method is entirely using own multislice code highly optimized for efficient simulation and gradient calculation.

In the authors' response to my point 5, I would point out that the sampling for ePIE is often expressed in terms of a percentage overlap of probes, but for a focused probe the overlap is essentially equivalent to the Nyquist sampling for the probe despite a Fourier transform with respect to probe position not being used. My question was really whether the authors' approach allowed for further relaxation of the sampling by having less probe overlap than needed by ePIE. If so, then this is a useful advantage of the authors' method.

Our current experience with the performance of the new methodology points indeed towards more relaxed requirements for the sampling. However, a general mathematical proof or reasoning has not been developed so far to confirm this theoretically. Therefore, we hesitate to mention this as a benefit in the manuscript, because it is not clear yet whether this observation is a fundamental feature or just applies to the cases dealt with here.

I agree that the discussion around my point 10 is some esoteric and not central to the work presented. It can be challenging to related discrete sampling events to a wave model, but my understanding is that the weak phase approximation (ie the truncation to the first term of the power series expansion of the phase) is regarded as a single scattering approximation (my recollection is that there is early work by Cowley on this). Higher order terms (ie the strong phase approximation) contain multiple scattering events, but in a thin sample. The term dynamical diffraction refers to multiple scattering with propagation (as included in a multislice calculation. I accept that the concept of multiple scattering in a thin sample (ie a uranium atom) is somewhat abstract. Perhaps the way this complication can be avoided is to simply assert that Wigner requires a multiplicative transmission function, and leave it at that.

We added to Supplementary Note 1: [...] (which uses a multiplicative transmission function).